# Climatic information archived in ice cores: impact of intermittency and diffusion on the recorded isotopic signal in Antarctica

Mathieu Casado[1], Thomas Münch[1], and Thomas Laepple[1]

[1]Alfred-Wegener-Institut Helmholtz-Zentrum für Polar- und Meeresforschung, Research Unit Potsdam, Telegrafenberg A45, 14473 Potsdam, Germany

**Correspondence:** Mathieu Casado (mathieu.casado@gmail.com)

**Abstract.** The isotopic signal ($\delta^{18}O$ and $\delta D$) imprinted in ice cores from Antarctica is not solely generated by the temperature sensitivity of the isotopic composition of precipitation but also contains the signature of the intermittency of the precipitation patterns as well as of post-deposition processes occurring at the surface and in the firn. This leads to a proxy signal recorded by the ice cores that may not be representative of the local climatic variations. Due to precipitation intermittency, the ice cores only record brief snapshots of the climatic conditions, resulting in aliasing of the climatic signal, and thus a large amount of noise which reduces the minimum temporal resolution at which a meaningful signal can be retrieved. The analyses are further complicated by isotopic diffusion, which acts as a low pass filter that dampens any high frequency changes. Here, we use reanalysis data (ERA-Interim) combined with satellite products of accumulation to evaluate the spatial distribution of the numerical estimates of the transfer function that describes the formation of the isotopic signal across Antarctica. As a result, the minimum time scales at which the signal-to-noise ratio exceeds unity range from less than a year at the coast to about thousand years further inland. Based on solely physical processes, we are thus able to define a lower bound for the time scales at which climate variability can be reconstructed from the isotopic composition in ice cores.

## 1 Introduction

Ice cores are key archives of past climatic conditions (Jouzel and Masson-Delmotte, 2010, and references therein) as a wide range of climatic parameters are recorded in the physical and chemical composition of the ice itself and of the air bubbles trapped within. Water isotopes are commonly used as a past temperature proxy due to the sensitivity of the isotopic composition to atmospheric temperature variations over the course of the water cycle (Dansgaard, 1964; Lorius et al., 1969). Antarctic ice cores have been used to reconstruct continuous high resolution temperature time series dating back $800\,000$ years (Petit et al., 1999; EPICA, 2004; Kawamura et al., 2017). Ice core water isotope data have also been used to compare rapid (e.g. Dansgaard–Oeschger) events between the Arctic and Antarctica (EPICA, 2006; Markle et al., 2017) or to provide a context for the recent climate change (Stenni et al., 2017). Even though the amount of water needed to analyse water isotopes is very small (Jones et al., 2017a), inhomogeneous deposition and diffusion together with the annual layer thickness limit the temporal resolution of the climatic signal that can be retrieved from isotopes. As a result, ice cores from high accumulation areas such as Coastal Antarctica (Morgan, 1985; Masson-Delmotte et al., 2003; Küttel et al., 2012; Vega et al., 2016; Caiazzo et al.,

2017; Goursaud et al., 2018) and West Antarctica (Markle et al., 2017) could be used to achieve up to seasonal resolution in temperature reconstructions, while ice cores from low accumulation areas such as the East Antarctic Plateau cannot be used to achieve a temporal resolution below decadal, or even multi-decadal (Petit et al., 1982; Ekaykin et al., 2002).

Due to how the signal is imprinted into the water isotopic composition in the ice of Antarctica, there are fundamental limits to the reconstruction of past temperatures based on water isotopes (Petit et al., 1982; Casado et al., 2017). For instance, before the precipitation forms near the deposition site, the isotopic composition of the atmospheric moisture keeps an imprint of all the fractionation processes that occurred since the water evaporated in the mid-latitudes (Craig and Gordon, 1965), including the subsequent condensation events that occurred while the air masses moved to high latitudes (Dansgaard, 1964). Moreover, the local climatic signal is only archived in the snow when there is a precipitation event (Steig et al., 1994; Werner et al., 2000; Sime et al., 2009) which introduces a bias and aliasing in the recorded signal (Laepple et al., 2011; Persson et al., 2011; Sime et al., 2011; Casado et al., 2013, 2018). Even after the deposition, in central Antarctica, the accumulation not only depends on the precipitation input (Genthon et al., 2015), but is also affected by blowing snow, moving the snow layers several times before they eventually settle (Picard et al., 2019), thereby redistributing and mixing snowflakes of different isotopic composition which leads to significant stratigraphic noise in the firn isotopic composition (Fisher et al., 1985; Ekaykin et al., 2002; Münch et al., 2016). In addition, isotopic diffusion in the firn acts as a low pass filter that erases part of the climatic signal (Johnsen, 1977; Johnsen et al., 2000; Gkinis et al., 2014; Laepple et al., 2018).

Overall, although the isotopic composition of precipitation in Antarctica is relatively well correlated with local temperature, both spatially and temporally (Landais et al., 2012; Stenni et al., 2016), the surface snow isotopic composition often is not (Touzeau et al., 2016), which suggests that it includes more processes than just precipitation as an input (Casado et al., 2018). In addition, comparisons of the statistical properties of the seasonal climate signal and the isotopic profiles in snow-pits have suggested that there is a large amount of noise (up to 90% of the total variance) in the input isotopic signal (Laepple et al., 2018). It is important to determine the origin of this noise, since post-deposition processes (wind blowing, sublimation/condensation at the surface, metamorphism, etc.) only affect the signal locally and are characterised by decorrelation lengths of the order of five to ten metres (Münch et al., 2017), while precipitation intermittency will have an effect over hundreds of kilometres, as suggested by an analysis of the spatial extent of the precipitation simulated by a regional climate model (Agosta et al., 2019).

While there is generally a high degree of confidence in ice cores being able to yield good results as an isotopic paleothermometer on long time scales (from multi-decadal to millennial), the climatic signal in the ice core on shorter time scales (seasonal to inter-annual) is often difficult to interpret, especially at low accumulation sites (Frezzotti et al., 2007). Although technical advances in analytical techniques (Jones et al., 2017b) have immensely improved the sampling resolution without any additional analytical costs, there is still a need to identify the minimum time scales at which it is possible to recover a meaningful climatic signal from the ice cores' isotopic composition. In other words, what is the theoretical lower limit for the time scale at which a meaningful climatic signal can still be reconstructed from water isotopes in ice cores? Here, we present

two simple modelling approaches involving virtual ice cores to identify the minimum resolution at which a climatic signal can be retrieved from the snow's isotopic composition in Antarctica at a predefined level of quality (signal-to-noise ratio). This approach will provide a theoretical understanding of the limitations of ice core records, particularly at low accumulation sites where the snow remains exposed for a long time before being buried. By combining approaches from Sime et al. (2011), Persson et al. (2011), and Laepple et al. (2018) we construct a forward model of virtual ice cores that includes (i) the climatic signal, (ii) precipitation intermittency, (iii) isotopic diffusion, and (iv) measurement noise. The relative importance of each of these four contributions is then compared in a conceptual spectral model to determine the lower bounds for the time scale above which a meaningful reconstruction is possible.

## 2 Data and methods

### 2.1 Forward model for ice core records

Following Sime et al. (2011) and Laepple et al. (2018), we developed a simple forward model that uses temperature and precipitation time series to simulate virtual ice cores. For each precipitation event, the model determines the corresponding amount of snow and applies an isotopic composition that is determined by the temperature during the event. The model also accounts for diffusion, and it outputs vertical depth series that can be dated to produce isotopic time series.

Temperature is converted to isotopic composition assuming a linear relationship with a constant slope of $0.46\,\%o\,°C^{-1}$ (Touzeau et al., 2016). We chose to use the same slope for all of Antarctica as this will allow us to distinguish between the noise generated by different distillation paths and the noise due to precipitation intermittency. A large range of slopes are reported in the literature, but in our case, the uncertainty on the value of the slope does not affect our main conclusions as both the input signal and the noise scale with the same coefficient. A different value for the slope only affects our results when considering the impact of measurement uncertainty: while the assumed isotopic signal and noise from intermittency scale with the slope, the measurement uncertainty stays constant and thus its relative impact changes.

Precipitation intermittency is computed as follows: every six hours (model time step) a new layer of snow is added to the previous stack with an isotopic composition determined by the temperature (first input of the model) and a thickness determined by the amount of precipitation (second input), converted to snow height using a snow density of $350\,kg\,m^{-3}$. This has two effects: (i) days without precipitation events will not leave any signature on the virtual ice core, and (ii) the statistical weight of the temperature on days with precipitation events is increased with the amount of precipitation. This yields an intermittent virtual core, whose total depth (in metres) is the product of the duration of the input signal ($a$) and the mean accumulation rate ($m\,a^{-1}$).

For the sake of simplicity, missing noise that could be introduced when the signal is archived (stratigraphic noise, metamorphism, etc.) is parametrised as a redistribution of the signal power across all frequencies prior to diffusion, equivalent to a random reshuffling of the signal in the time domain (Münch et al., 2016). In practise, this is implemented by adding temporally

independent white noise to the intermittent virtual core and renormalising the variance of the total signal (intermittent virtual core + white noise) to the variance of the original signal (intermittent virtual core only) (Laepple et al., 2018). The added white noise is controlled by two parameters: (i) the relative amount of noise compared to the input signal (0 to 100%) and (ii) the resolution at which the noise impacts the signal (from 1 to 10 cm). The noise module in the model is used to assess the impact of additional noise sources on precipitation intermittency and isotopic diffusion.

Diffusion is applied using the classical isotopic diffusion scheme (Johnsen, 1977; Johnsen et al., 2000; Gkinis et al., 2014) of convolving the depth series with a Gaussian kernel (Johnsen et al., 2000) following Fick's law. It is characterised by a depth-dependent diffusion length (Laepple et al., 2018) that is computed for each site based on the local temperature, accumulation rate, atmospheric pressure, and the snow density. We model the snow density profiles using the Herron and Langway model (Herron and Langway, 1980) assuming a constant surface density of $350 \, \mathrm{kg \, m^{-3}}$ and setting the temperature of each site to the ERA-Interim grid point value. Atmospheric pressure is kept constant at 650 mbar. The impact of both the constant atmospheric pressure and surface snow density on the diffusion length is minimal and allows for a straightforward comparison of different sites.

The virtual records (intermittent virtual core and diffused virtual core) are block-averaged to create a 1 cm vertical resolution, similar to what can be achieved with manual sampling of ice cores. The virtual ice cores are perfectly dated by tagging the formation date and time to each layer. During the block-averaging to 1 cm, we also block-average the date tags to obtain the average age of each 1 cm layer. The perfect dating can be used to compare the original climatic signal to the generated virtual cores, in an optimistic case (without age uncertainty, equivalent to every snow layer being perfectly dated). To do so, we do a linear interpolation of the virtual ice core to the original date coordinates of the input data. Indeed, the original climatic depth series typically shows a rather poor correlation with the generated virtual cores as their respective depth axes move quickly out of phase due to the large interannual variability in precipitation, which creates years accounting for thicker/thinner layers when the amount of precipitation is large/small. In contrast, a perfect record would only contain the climatic signal and produce the same layer thickness each year. The perfect dating enables to synchronise the virtual cores' time series on the climatic signal in order to provide an upper bound of how meaningful a reconstruction be would be ignoring dating issues. In most of the manuscript, a more realistic case study is presented for which a constant accumulation rate is considered between two tie points (39 years in the manuscript, with sensitivity tests presented in Supplementary Materials S3).

## 2.2 Input time series and correction

As inputs we use a 39-year (1979–2018) time series of 2 m air temperature (T) and total precipitation (P), both from ERA-Interim re-analysis (Dee et al., 2011) at a temporal resolution of six hours and a spatial resolution of approximately 80 km (T255 spectral truncation).

The ERA-Interim temperature data provide good approximations of the spatial and temporal variations of the temperature observed in *in-situ* data from Antarctica (Genthon et al., 2013; Medley et al., 2013; Jones and Lister, 2015). However, compared to satellite products and *in-situ* ice core records, the ERA-Interim data overestimates the amounts of total precipitation by 50 to 95% (Arthern et al., 2006; Thomas et al., 2017). This is to be expected as precipitation amounts do not directly contribute to the local accumulation in Antarctica, especially in Central Antarctica where up to 90% of the local accumulation can be blown away by wind (Picard et al., 2019) and more than 10% of the total surface mass balance can be associated with sublimation and condensation (Genthon et al., 2017). Nevertheless, precipitation occurrence tends to correlate well with *in-situ* snowfall events in the interior of Antarctica (Medley et al., 2013; Libois et al., 2014), and with ice core records from Antarctica (Sime et al., 2011). This supports the use of ERA-Interim precipitation since a well-captured precipitation variability is needed to realistically model the precipitation intermittency.

However, since the diffusion length depends on the amount of accumulation, we need to compensate for the difference between precipitation and accumulation. This is achieved by applying an individual linear correction at each grid point of the reanalysis product. The correction matrix was generated using satellite data of snow accumulation (Arthern et al., 2006), that were corrected before to match the accumulation obtained in ice core records (Thomas et al., 2017) with the following procedure. For the virtual cores to have the same accumulation as actual ice cores, we used a reference accumulation rate for the years from 1960 to 2016 from a recently established database of regional Antarctic snow accumulation from ice core records over the past 1000 years (Thomas et al., 2017). We selected all the ice cores sites with accumulations ranging from 20 to $400 \, \mathrm{kg \, m^{-2} \, a^{-1}}$ that had overlap with the ERA-Interim time series (in total 71). The accumulation range upper limit ($400 \, \mathrm{kg \, m^{-2} \, a^{-1}}$) was chosen to be representative of the low accumulation rates of the deep ice core sites (in general $<100 \, \mathrm{kg \, m^{-2} \, a^{-1}}$) where the results are more sensitive to the use of an accurate accumulation rate. We then did a spatial linear regression between the satellite derived accumulation (Arthern et al., 2006) for these 71 sites and the ice cores observations, and used the produced regression to calibrate the satellite data of snow accumulation on the ice core accumulation rates. Finally, we interpolated the corrected satellite product to ERA-Interim grid, and used the corrected satellite product as a reference for the accumulation, normalising the precipitation amount of ERA-Interim to match this reference.

The impact of this correction was then assessed using the uncorrected ERA-Interim amount of precipitation (See Supplementary Material S2). This affects the results locally as values of accumulation will not match reality, thereby changing the diffusion length while the modelled impact of precipitation intermittency and stratigraphic noise for a given accumulation amount remain unaffected.

In addition, we use the millennial CMIP5 climate model simulations to compare our results with longer time series than the ones produced by ERA-Interim. We use the past1000 simulations from eight General Climate Models (GCM), namely BCC-CSM1-1, CCSM4, CSIRO-Mk3L-1-2, FGOALS-gl, GISS-E2-R, IPSL-CM5A-LR, MIROC-ESM, MRI-CGCM3 that cover the last 1000 years and include the historical solar and volcanic forcing (Bothe et al., 2013).

## 2.3 Evaluating the signal-to-noise ratio in the spectral domain

In a second modelling approach, we employ a method in the frequency domain (spectral method) to evaluate the ice core signal as a combination of (i) the climatic signal, (ii) noise linked to precipitation intermittency, (iii) additional noise of unknown origin, (iv) a low pass filter due to isotopic diffusion, and (v) measurement noise. The purpose of this spectral approach is to produce time-scale dependent signal-to-noise ratios (SNR) that allow estimations of the time scales at which ice cores will be correlated with the climatic signal. We make use of the outputs of the forward model (Sect. 2.1) to parameterise this conceptual spectral model.

In the spectral domain, the noise added by precipitation intermittency originates from sub-sampling the climatic signal (dominated by the seasonal cycle) which in turn leads to aliasing as only the temperatures during precipitation events are recorded. Empirically, precipitation events are largely random in Antarctica (Genthon et al., 2003; Rémy and Parrenin, 2004; Turner et al., 2019). During the aliasing of a signal by random sub-sampling, the superimposed noise is white (Thomson and Robinson, 1996). The whiteness of the precipitation intermittency noise is confirmed by numerically examining the impact of precipitation intermittency using ERA-Interim data (Fig. 4). Thus, throughout this manuscript, we use this approximation and consider the added noise as white.

To evaluate the extent to which the climatic signal is preserved in the ice core record as a function of time scale, we assess the minimum time scale $\tau$ at which the SNR reaches a value of 1, denoted in the following as the *signal retrieval time scale*. In any proxy record containing a climatic signal and noise, the SNR and the correlation between the record and the climatic signal are linked via

$$r^2 = \frac{\text{SNR}}{1 + \text{SNR}}. \tag{1}$$

As a result, the signal retrieval time scale will correspond to a correlation with the climate signal of $r = \sqrt{0.5} \sim 0.71$.

The SNR can be defined in two ways. First, one can analyse the SNR at a specific frequency after filtering the ice-core time series with a narrow bandpass filter, in which case we refer to the signal retrieval time scale as $\tau_b$, where the subscript $b$ stands for bandpass. Second, and more commonly, the ice-core time series results from averaging a higher-resolution record to a fixed temporal resolution, either by discrete sampling in the depth domain or by block-averaging the dated record to a specific resolution, e.g. annual or decadal resolution. In this case, we refer to the signal retrieval time scale as $\tau_a$, where the subscript $a$ stands for averaging.

Formally, $\tau_b$ is given by the critical frequency $f_b = 1/\tau_b$, for which the direct ratio of the signal and noise spectra reaches 1,

$$\text{SNR}(f_b) = \mathcal{S}(f_b)/\mathcal{N}(f_b) = 1 \tag{2}$$

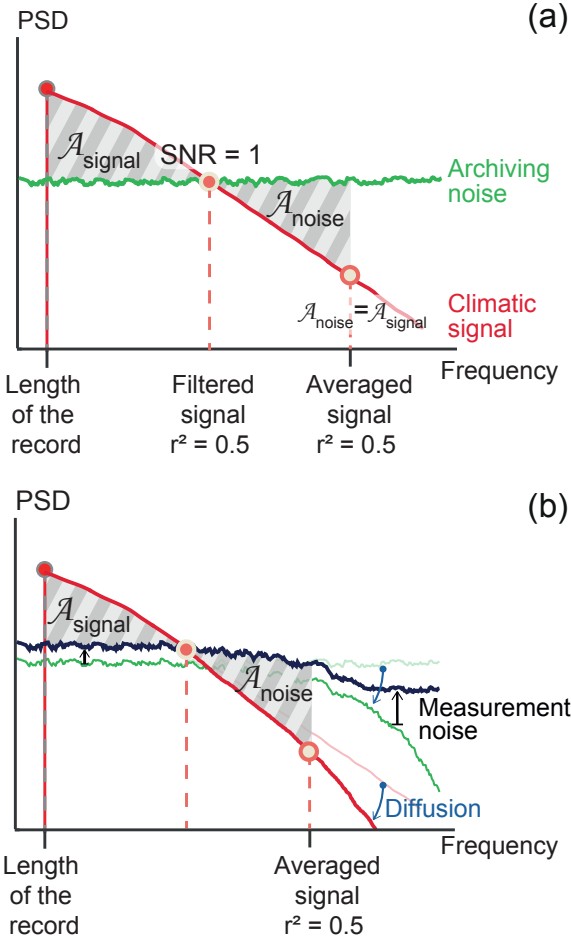

**Figure 1.** Schematic of the signal-to-noise ratio (SNR) estimate using the power spectral density (PSD): (a) idealised PSD of the climatic signal (red) and of the archiving noise (green) which consists of noise induced by precipitation intermittency and stratigraphic noise; (b) impact on the same PSDs of the effect of diffusion (light colours: before diffusion, dark colours: after diffusion) and of measurement noise (black).

where $\mathcal{S}$ and $\mathcal{N}$ are the power spectral densities (PSDs) of the signal and the noise. To obtain $\tau_a$, the PSDs of the signal and noise have to be integrated to that critical frequency $f_a = 1/\tau_a$, for which the ratio of the *integrated* spectra reaches 1,

$$195 \quad \mathrm{SNR}(f_a) = \frac{\int_{1/L_R}^{f_a} \mathcal{S}(\nu)d\nu}{\int_{1/L_R}^{f_a} \mathcal{N}(\nu)d\nu} = 1 \tag{3}$$

where $L_R$ is the length of the record (either in years or in metres). Graphically, this is given by the ratio of the area representing the signal excess $\mathcal{A}_{signal}$ to the area representing the noise excess $\mathcal{A}_{noise}$ (Fig. 1a). Note that due to the Shannon-Nyquist theorem, the actual resolution of the record would have to be $\tau_a/2$ in order to obtain an SNR of 1 at the frequency $1/\tau_a$. However, here we use $\tau_a$ for the signal retrieval time scale in order to ease the comparison with $\tau_b$. If the signal is redder than

the noise, $\tau_a$ will be smaller than $\tau_b$.

With regard to precipitation intermittency, assuming that the noise related to the aliasing of the climatic signal is white, we can estimate the amount of noise contained at the interannual to decadal scales using ERA-Interim data and then apply it to lower (centennial and millennial) frequency ranges. We then compare the amount of climatic signal (provided independently) 205    to the constant white noise due to precipitation intermittency and estimate the frequency at which the SNR reaches a value of 1. In addition to the noise induced by precipitation intermittency, several other sources of noise can influence the signal recorded by ice cores such as stratigraphic noise (Fisher et al., 1985), and we combine all types of noise in this method to the collective term of *archiving noise* (Fig. 1a).

Adding the impact of diffusion results in a convolution of both the signal and the archiving noise with the diffusion transfer function (Fig. 1b) and thus does not directly affect the SNR, which is also reflected by the fact that diffusion can be inverted ("back diffusion") (Münch and Laepple, 2018). However, in practice, additional measurement noise is added to the signal after the diffusion has taken place, which limits the potential of back-diffusion. We include this effect in the estimation of the signal retrieval time scales by adding a measurement noise offset to the initial archiving noise after diffusion (Fig. 1b). This 215    additional noise reduces the area of excess signal and increases the area of excess noise, thereby reducing the frequency at which the correlation between an ice core record and the climatic signal reaches $r^2 = 0.5$.

## 2.4    Comparison with PSD from snow pits in Antarctica

We compare our results with snow-pit data from three deep ice-core sites in Antarctica (West Antarctic Ice Sheet, WAIS; EPICA Dronning Maud Land, EDML; and Dome C). These three sites were chosen to illustrate a large range of climatic 220    conditions for which we expect different signal retrieval time scales: WAIS with an expected close-to-annual signal retrieval (Jones et al., 2018), EDML as an intermediate site for which it has been shown that only decadal signals and longer can be retrieved (Münch and Laepple, 2018), and Dome C as a site where we expect to have even longer signal retrieval time scales (Petit et al., 1982).

The WAIS site is located in West Antarctica (79.5°S, 112°W) and the dataset selected for this study includes two 30 m vertical profiles of isotopic composition, including the WAIS Divide ice core (WAIS Divide Project members, 2013) and a shallow core obtained in 2006 (Jones et al., 2018). Considering the local accumulation (around $220\,\text{kg}\,\text{m}^{-2}\,\text{a}^{-1}$ w.e.), this depth range covers more than 39 years of snow accumulation, and thus encompasses the period covered by ERA-Interim in this study. From the EDML site located in East Antarctica (75.0°S, 0.1°E) we use here two 3.5 m profiles of average isotopic 230    composition obtained from trench studies (Münch et al., 2017) as well as two 12 m cores (B41 and B50; Laepple et al., 2018). Finally, from the East-Antarctic site of Dome C (75.1°S, 123.3°E), we include three snow pits covering between 1 m and 3.5 m depth (Touzeau et al., 2016; Casado et al., 2018).

## 3 Results

### 3.1 Illustrating the methodological approach

In order to illustrate the methodological approach and the results of the forward model, we choose the EDML site near Kohnen Station, for which a large number of snow pits are available (Münch et al., 2017; Laepple et al., 2018).

A first virtual core is generated for the pure climate signal (Fig. 2b) which corresponds to a perfect record as each day is archived and no information is lost. The climatic signal at EDML is dominated by the seasonal cycle, as for most sites in 240  Central Antarctica, which leads to a large peak in the PSD at the frequency that corresponds to the local accumulation and a smaller peak corresponding to the second harmonic (Fig. 2c).

Since water isotopes create an archive of the temperature conditions only during precipitation events, many days will not be recorded. In addition, large precipitation events will lead to thicker layers of snow which in turn have a stronger statistical im-245  pact on the overall signal recorded in the ice core (Fig. 2d). These effects are included in the precipitation intermittency virtual core (Fig. 2e). Here, the amount of variance is reduced, as the precipitation events in winter are often associated with warmer than average conditions, which leads to an under-representation of the coldest conditions and a warm bias in the isotopic record (Noone et al., 1999; Casado et al., 2018). As a result, the difference between the precipitation-weighted temperature and the actual temperature is larger in winter than in summer (Fig. S4). Indeed, the amount of lost variance (the difference between the 250  variances of the intermittent and the climatic virtual cores) is throughout Antarctica positively correlated with the difference between the mean value of the intermittent virtual core and the climatic core ($r^2 = 0.34$, n = 12128, p < 0.05). This suggests that part of the variance reduction is related to the under-sampling of the colder winter conditions (Fig. S4). Overall, the total amount of variance preserved in the intermittent virtual core ranges from 30 to 100% of the amount of variance observed in the climatic signal. However, the PSD of the intermittent virtual core is very different from the climatic signal, and the large 255  amount of variance at the frequency equivalent to 1 year in the climatic signal is reduced, since precipitation intermittency redistributes the very strong seasonal signal across all frequencies, as can be seen from the example of EDML (Fig. 2f).

Our modelling approach produces profiles of isotopic composition which can be plotted either against depth or against time. Analysing the depth series, we observe no correlation between the intermittent and the climatic virtual cores, since the seasonal 260  cycles are out of phase due to the interannual variations in the amount of precipitation. Analysing the time series, assuming that each layer is perfectly dated (perfect dating assumption), we obtain for EDML a correlation of $r = 0.85$ ($p < 0.05$) between the virtual core and the climate signal.

Accounting for diffusion reduces the variance (Fig. 2h) mainly due to the damping of high frequency variations. At EDML, 265  the diffusion low-pass filter starts to have a strong effect at frequencies corresponding to length scales smaller than the local accumulation rate, i.e. at the interannual scale. Under the perfect dating assumption, the diffused core is correlated to the

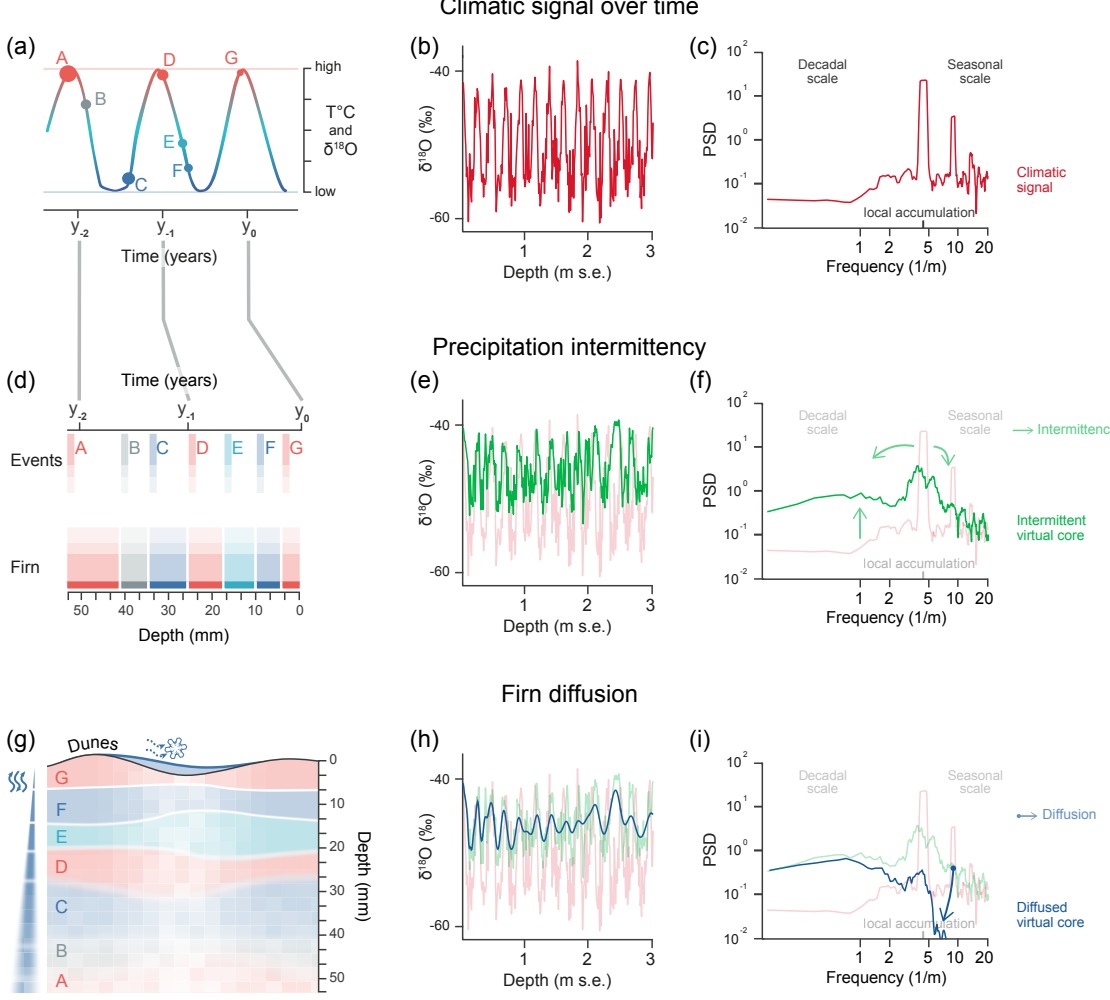

**Figure 2.** Description of the archival processes included in the forward model that lead to a loss of signal in the snow isotopic composition. Examples are for EDML: (a) idealised temperature time series with rare precipitation events (capital letters A to G); (b) climatic virtual core: actual temperature time series converted to an isotopic profile for the case of constant daily precipitation; (c) power spectral density of the climatic virtual core in (b); (d) schematic illustrating the impact of precipitation intermittency on the layering of the isotopic profiles; (e) intermittent virtual core: isotopic composition after precipitation intermittency has affected the signal; (f) power spectral density of the intermittent virtual core in (e); (g) schematic illustrating the impact of snow redistribution and isotopic diffusion on the snow layering; (h) diffused virtual core: isotopic composition after precipitation intermittency and diffusion have impacted the signal, (i) power spectral density of the diffused virtual core in (h).

climatic signal with $r = 0.22$ ($p < 0.05$), mainly due to the most recently deposited near-surface layers that have not been

diffused yet. This is visible in the PSD of the diffused core as a peak remaining near the frequency that corresponds to the annual accumulation (Fig. 2i).

## 3.2  Outputs of the forward model across Antarctica

By producing similar virtual cores for each grid point of the ERA-Interim reanalysis product, we can illustrate the impact of the archival processes on the signal by comparing the correlation of the virtual cores of each site with the climatic signal under a perfect dating assumption (Fig. 3). Precipitation intermittency alone (Fig. 3a) only slightly reduces the correlation of the full time series (mean correlation across Antarctica: $r = 0.88$, $p < 0.05$). As the seasonal cycle clearly dominates the signal by roughly two orders of magnitude in the frequency range covered by ERA-Interim (Fig. 2c), the correlation is reduced at a large number of interior sites when applying a two-year running mean filter (henceforth referred to as interannual low-pass filter), which illustrates the aliasing effect due to precipitation intermittency at interannual and decadal scales (Fig. 3c).

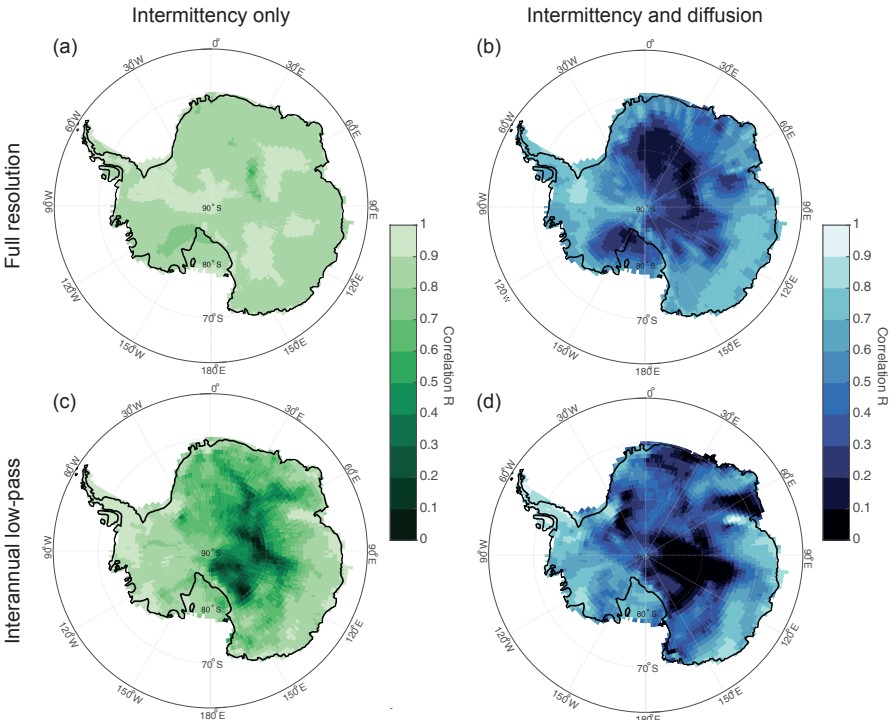

**Figure 3.** Maps showing the correlation between the climatic signal and perfectly dated virtual cores: (a) and (c) precipitation intermittency only (green); (b) and (d) precipitation intermittency and diffusion (blue). The top panels (a) and (b) present the correlation using the complete time series, while the bottom panels (c) and (d) show the correlation for the time series smoothed with a two-year running mean filter.

After diffusion, the correlation between the virtual core and the climatic signal drops on the East Antarctic Plateau, Marie Byrd Land, and the Ross Ice Shelf (Fig. 3b). Part of the remaining correlation is due to remnants of the seasonal cycle that

have been preserved (mean correlation across Antarctica of $r = 0.50$). By filtering out any signal below the interannual scale, large areas exhibit a drop in correlation, particularly in the interior (Fig. 3d). In areas that exhibit a drop in correlation while the forward model shows significant correlation with the climate signal at sub-annual resolution (up to $r = 0.9$), there is no power of reconstruction, because the artificial signals at interannual scales due to precipitation intermittency make it impossible to

retrieve any climatic signal.

The perfect dating assumption corresponds to an ideal case for which each layers of snow is dated. This is in most cases unrealistic, for instance, in the case of EDML, this corresponds to dating snow layers every 16 days on average. In the following part of the manuscript, we include a more general case for ice core records interpretation for which the conversion from depth

to age is done by considering a constant accumulation rate in between two tie points. In this study, we use the full dataset of ERA-interim (39 years) to calculate the accumulation, leading to a dating interval of roughly 40 years, over which the average accumulation rate is used to convert the results from the depth domain to the time domain. This value is realistic considering that the most common tie points for ice core records (volcanic ashes layers) rarely occur more often than this (Gautier et al., 2016). Sensitivity tests on the impact of the dating interval on the results are presented in Supplementary Materials S3.

**3.3   Impact of intermittency on long time scales**

In order to investigate the effect of precipitation intermittency for time scales relevant in ice-core studies, we extend our results up to centennial and millennial time scales, in the case where the dating relies on constant accumulation rate for in between tie point with dating interval of roughly 40 years. As the ERA-Interim time series input we use only covers 40 years, we do this by making use of (i) the approximation that the noise generated by precipitation intermittency is white (see Sect. 2.3), and (ii)

general assumptions on the spectrum of the climatic signal.

We estimate the noise level added by precipitation intermittency from the difference between the power spectral densities of the intermittent virtual core and the climatic signal virtual core over the interannual and decadal scales (more specifically, for frequencies below $3/2 \, \mathrm{a}^{-1}$, hatched area in Fig. 4). As the noise is white, we generalise this level to longer time scales (see

the dashed line in Fig. 4). For instance, in the case of EDML, we obtain a white noise level of $0.59 \, \%o^2 \, \mathrm{m}$ using the difference between the intermittent virtual core (green curve) and the climatic virtual core (red curve). For the period covered by ERA-Interim, the signal strength never reaches this noise level at EDML, except for the frequency associated with the seasonal cycle (Fig. 4). In consequence, the SNR-based correlation at interannual scales between the intermittent virtual core and the climatic signal will be below $r^2 = 0.5$, and specifically the time series correlation obtained by comparing the virtual cores at EDML is

$r^2 = 0.15$.

To obtain an input climatic signal for longer time scales, we use in a first step the spectra of the 1000-year long forced simulations from the CMIP5 model ensemble. For this, we first produce the PSDs of the temperature data for the last 1000 years for each grid point of the eight CMIP5 models from the past1000 runs. We then resample this field of temperature PSDs

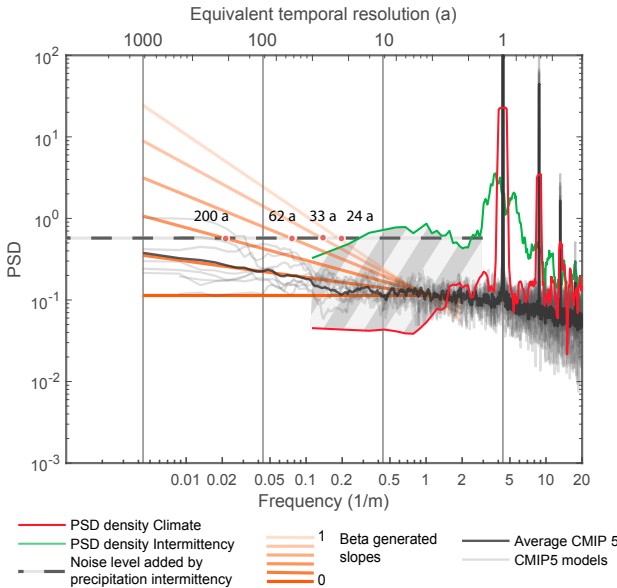

**Figure 4.** Comparison of the amount of noise generated by precipitation intermittency and different hypothesis for the climatic signal for the site of EDML: PSD generated by the forward model for the climatic signal (red) and the precipitation intermittency (green) virtual cores; area where the noise level added by precipitation intermittency is calculated (hatched zone) and noise level threshold (hatched line); and several hypothesis for the climatic signal input over 1000 years. The signal retrieval time scales $\tau_b$ (see Sect. 2.3) are given as the intersection of the noise level and the climatic signal inputs.

to the ERA-Interim grid and convert the spectra to $\delta^{18}O$ units (see Sect. 2.1). The results show (grey lines in Fig. 4) that while some individual models predict a sufficiently strong signal that exceeds the noise created by precipitation intermittency, the SNR for the average signal remains below 1.

        Next, we consider alternative assumptions to GCMs regarding the climate variability on longer time scales, which might
more accurately represent the amount of variance observed in ice cores. One of the simplest parametrisations to describe the observed climate variability over a large range of time scales is to assume a power-law relationship for the PSD of the signal $\mathcal{S}(f)$ (Huybers and Curry, 2006; Lovejoy and Schertzer, 2013),

$$\mathcal{S}(f) \propto f^{-\beta}, \tag{4}$$

        where $\beta$ is the scaling exponent. For the CMIP5 model ensemble, the scaling exponent obtained at EDML is $\beta = 0.2$, while
EDML ice core records for the last 1000 years indicate a scaling exponent of around $\beta = 0.6$ after correcting for local non-climate variability (Münch and Laepple, 2018). Here, we investigate $\beta$ between 0 and 1 to cover the range of reasonable scaling behaviours of temperature.

When we analyse the amount of signal at a specific single frequency (for instance, as it is commonly done to evaluate the solar cycles), an SNR of 1 is reached at a temporal resolution of $\tau_b$ (Sect. 2.3). In the best case scenario, as no other effects than precipitation intermittency are considered, this signal retrieval time scale varies for an ice core drilled at EDML from a value of more than 1000 years, if we characterise the climatic conditions by a power law with a slope of $\beta = 0$ or 0.2, to 24 years for a power-law slope of $\beta = 1$ (Table 1).

**Table 1.** Signal retrieval time scales at EDML after precipitation intermittency for the cases of band-pass filtering at a specific frequency ($\tau_b$) and of block-averaging to a certain resolution ($\tau_a$), both as a function of the input climate spectrum scaling exponent $\beta$.

| $\beta$ | 0 | 0.2 | 0.4 | 0.6 | 0.8 | 1 |
|---|---|---|---|---|---|---|
| $\tau_b$ (a) | / | / | 200 | 62 | 33 | 24 |
| $\tau_a$ (a) | / | / | 91 | 17 | 7.4 | 3.9 |

Usually, ice cores records consist of block-averages of the vertical profile, at a given resolution (either by measuring long bars of ice as a block, or by averaging samples from measurements done at a finer scale). For this approach, the time scales at which the SNR will reach a value of 1 are given by the signal retrieval time scale $\tau_a$ after block-averaging (Sect. 2.3). For a time series of 1000 years at EDML, $\tau_a$ is smaller than $\tau_b$ with a value of $\sim 4$ years for $\beta = 1$ (Table 1).

We generalise these results to all of Antarctica and present maps of the signal retrieval time scales $\tau_a$ at which the climate signal is preserved in an ice-core record after the impact of precipitation intermittency (Fig. 5). For this, we present maps of $\tau_a$ for values of power-law slopes of $\beta = 0.2$ (small scaling predicted by GCM and as estimated for climate variability in firn cores from the WAIS region; Münch and Laepple, 2018), $\beta = 0.6$ (best guess from isotope data from East Antarctica over the last millennium; Münch and Laepple, 2018), and a slightly higher value of $\beta = 0.8$, which is within the range of scaling exponents expected for decadal-to-centennial scale variations (Zhu et al., 2019). For a value of $\beta = 0.2$, for most of Antarctica, the time scale $\tau_a$ is larger than 1000 years, meaning that the amount of signal is too low to be visible in the ice core in most sites. Only in coastal areas can time scales below 50 years be obtained. For a value of $\beta = 0.6$, the time scales range from one year in coastal areas to 1000 years for special areas of the interior (e.g. Ellsworth Land and Victoria Land). For a value of $\beta = 0.8$, i.e. assuming more low frequency climate variability, the time scales are globally reduced. In both cases, the spatial pattern cannot be entirely explained by the amount of accumulation (Fig. 5 and Fig. S1): while the low-accumulation areas of the East Antarctic Plateau have large values for the signal retrieval times scales $\tau_a$ (from 10 to 500 years), the largest values of $\tau_a$ are however found for the regions around Ellsworth Land where the amount of accumulation is much larger (see Supplementary Material).

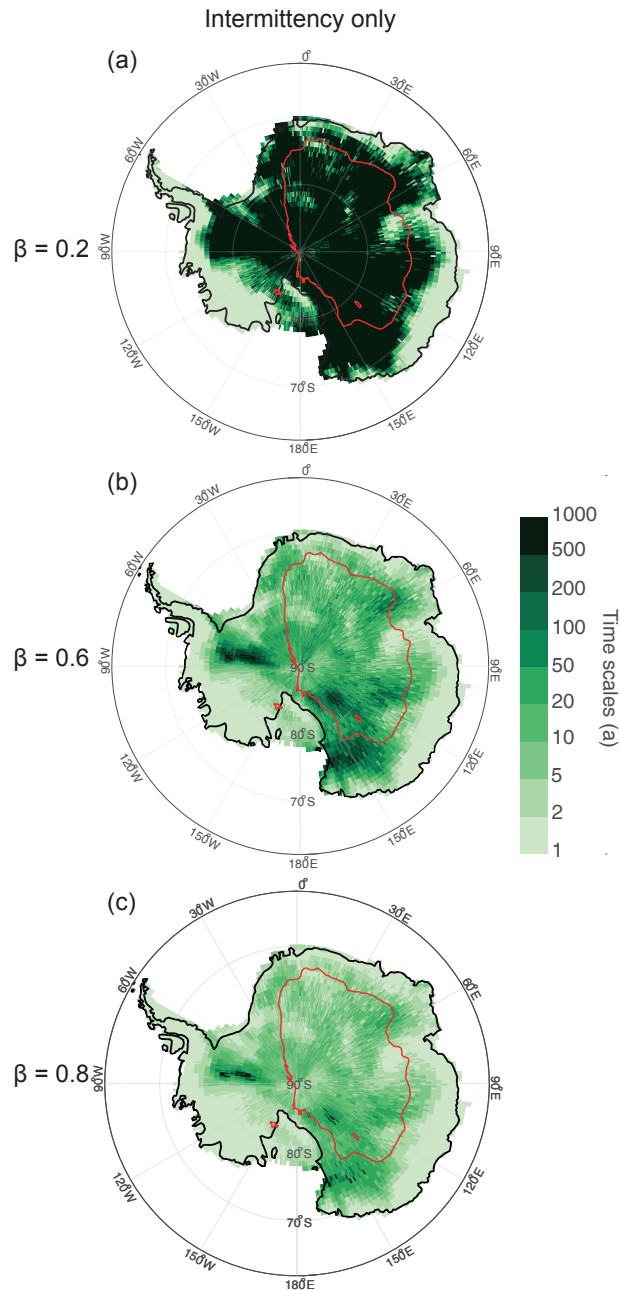

**Figure 5.** Maps of the signal retrieval time scales $\tau_a$ after block-averaging for the case of precipitation intermittency only and for climate spectrum scaling exponents of (a) $\beta = 0.2$, (b) $\beta = 0.6$ and (c) $\beta = 0.8$. The red line marks the contour of an accumulation of 25 cm w.e., see also Fig. S1.

## 3.4 Impact of diffusion

Isotopic diffusion continues to affect the isotopic signal after snow deposition has occurred; as a result, its impact increases with depth in the firn column. To illustrate the effect of diffusion on our time-scale estimates and compare it to the impact of precipitation intermittency, we use diffusion length values determined at the lock-in depth (for simplicity assumed here to be at $100\,\mathrm{m}$ depth).

**Table 2.** EDML signal retrieval time scales $\tau_a$ after block-averaging accounting for precipitation intermittency, diffusion, and measurement noise.

| $\beta$ | 0 | 0.2 | 0.4 | 0.6 | 0.8 | 1 |
|---|---|---|---|---|---|---|
| $\tau_a$ (a) | / | / | 143 | 23 | 9.2 | 4.6 |

As before, we assume climate input signals characterised by scaling exponents $\beta$ varying from 0 to 1 and compare these to a given noise level. We apply the diffusion transfer function both to the input signal and to the white noise spectrum generated by precipitation intermittency. Additional noise, which impacts the signal after diffusion such as measurement noise, will limit the ability to back-diffuse the ice core signal. We take this into account by adding a measurement noise level of $0.1\,\%o^2$. As above, we calculate the signal retrieval time scales $\tau_a$ for which the $\mathrm{SNR} = 1$ after block-averaging, with the noise spectrum now including both diffused white noise linked to precipitation intermittency and pure white noise linked to measurement uncertainty.

Diffusion and the additional measurement noise mainly affect areas with low accumulation where diffusion is more important (Fig. 6), which leads to an increase in the signal retrieval time scales mainly in the interior, such as at EDML where the values are larger especially for smaller values of $\beta$ (Table 2).

## 3.5 Comparison with snow-pit *in-situ* measurements

We compare the PSDs of the simulated isotopic profiles that include precipitation intermittency and diffusion to the PSDs of observational snow-pit isotope data obtained from sites all across Antarctica that exhibit a wide range of accumulation and temperature conditions (Fig. 7). As illustrated by Laepple et al. (2018), observations from snow pits lack any clear periodicity, particularly at the frequency associated with the seasonal cycle. In contrast, our model maintains some additional power at the frequency associated with the local accumulation rate, even when accounting for precipitation intermittency and diffusion (see the blue vs. black curves in Fig. 7). In addition, the model output shows reduced variance compared to the observations at the lowest frequencies (between $0.1\,\mathrm{m}^{-1}$ and the frequency associated with the local accumulation rate).

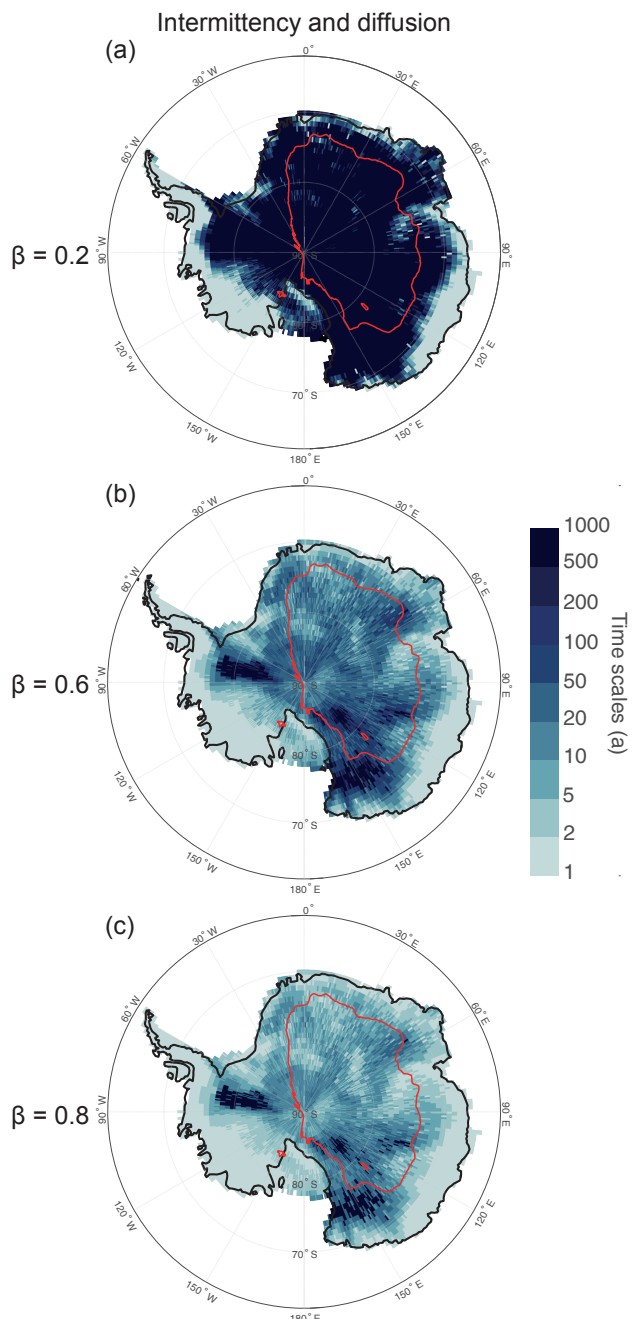

**Figure 6.** Maps of the signal retrieval time scales $\tau_a$ after block-averaging for the case of precipitation intermittency, diffusion and measurement error (0.1 ‰$^2$) affecting the signal and for climate spectrum scaling exponents of (a) $\beta = 0.2$, (b) $\beta = 0.6$ and (c) $\beta = 0.8$. The red line marks the contour of an accumulation of 25 cm w.e., see also Fig. S1.

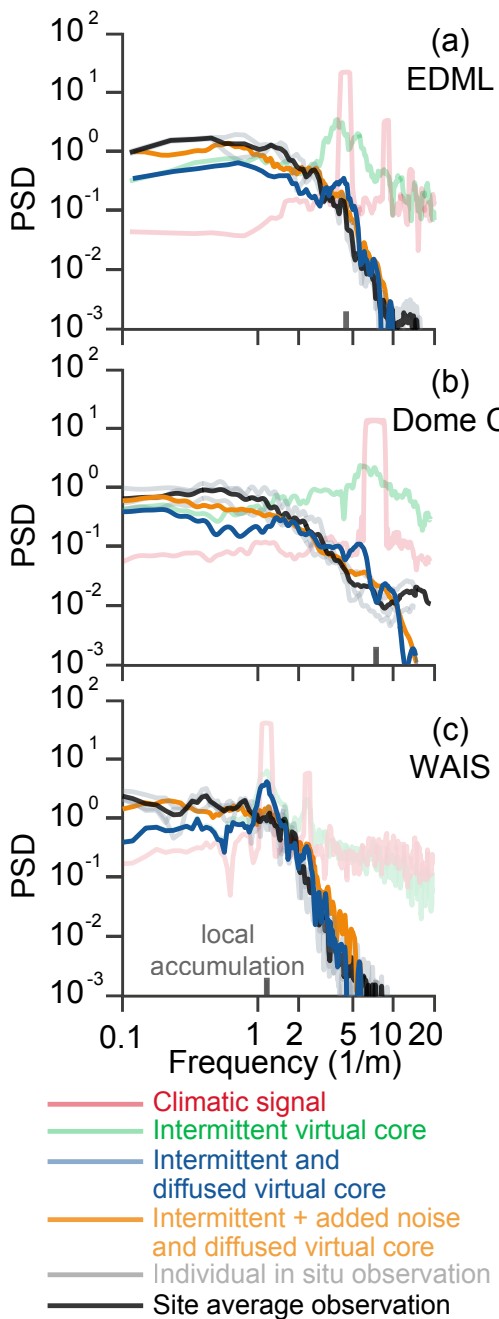

**Figure 7.** Power spectral density of the virtual ice cores compared to those of isotopes profiles from snow pits for (a) EDML, (b) Dome C, and (c) WAIS: climatic signal (light red), intermittent virtual core (light green), intermittent and diffused virtual core (light blue), intermittent + added noise and diffused virtual core (orange), snow pits from the sites (individual: grey, average: black).

Laepple et al. (2018) have shown that in order to generate accurate PSDs of snow pits in Antarctica, up to 90% of the total variance of the input signal before diffusion needs to be white noise. The patterns observed here correspond to a lack of noise compared to their estimates, which is expected as long as we do not account for stratigraphic noise (see Discussion). We can produce modelled profiles that better reproduce the PSD of observations for the three sites described here by converting more of the signal to white noise. In order to obtain the best fitting PSDs, the added noise accounts for 60% of the total variance at EDML and affects the virtual core at a resolution of 5 cm. At Dome C, the parametrisation of added noise that yields the best fit is 80% of the total signal at a resolution of 2 cm. At WAIS, the corresponding noise level is 80% of the total signal at a resolution of 10 cm.

## 4 Discussion

### 4.1 Impact of the results on the interpretation of ice core records

Ice core isotopic composition is traditionally used as a temperature proxy. For sites with very low accumulation such as Vostok, Dome C, or Dome F, where the oldest ice core records have been obtained (Petit et al., 1999; EPICA, 2004; Kawamura et al., 2017), temperature records are typically retrieved at centennial or decadal scales (EPICA, 2006). For instance, in the Dome C ice core, the 55 cm sampling rate and the varying accumulation yielded a temporal resolution of 15 to 30 years during the last Glacial Period. Here, we suggest that the temporal resolution of the time series obtained from ice core records should not be based just on the sampling rate of the ice core. Our results cast doubts on the amount of the climate signal that can be retrieved from very high resolution records (below annual), even if the climatic signal is back-diffused. The unexpectedly large impact

**Table 3.** Signal retrieval time scales $\tau_a$ (SNR $= 1$ after block-averaging) for the main Antarctic ice core sites including the impact of precipitation intermittency, diffusion and measurement noise and for three assumed climate spectrum scaling exponents.

| Ice core | Scaling as GCM | Prescribed scalings | |
| --- | --- | --- | --- |
| sites | $\beta \approx 0.2$ | $\beta = 0.6$ | $\beta = 0.8$ |
| Dome C | $> 1\,000$ | 27.8 | 10 |
| EDML | $> 1\,000$ | 23 | 9.2 |
| Vostok | $> 1\,000$ | 30 | 11 |
| Dome F | $> 1\,000$ | 12 | 4.6 |
| Dome A | $> 1\,000$ | 29 | 11 |
| South Pole | $> 1\,000$ | 15 | 6.5 |
| RICE | 2.3 | 1.1 | 0.81 |
| TALOS | $> 1\,000$ | 100 | 27 |
| WAIS | 9.0 | 1.8 | 1.3 |
| Law Dome | $< 0.50$ | 0.66 | 0.50 |

of precipitation intermittency in the form of white noise is masking much of the high frequency variability. In agreement with previous studies, our results suggest that climatic signals at time scales below decadal to multi-decadal cannot be recovered from ice cores collected on the East Antarctic Plateau (Petit et al., 1982; Ekaykin et al., 2002).

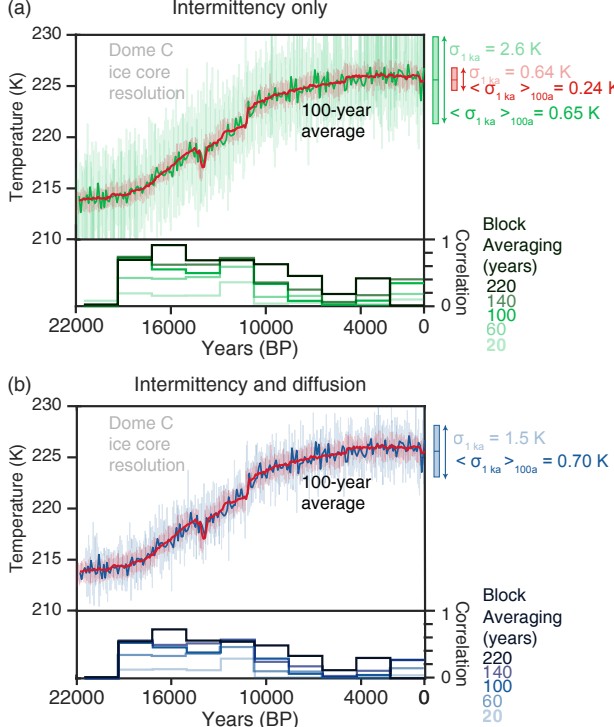

**Figure 8.** Application of the forward model to the temperature time series of the last deglaciation obtained from the Trace21k model simulation for Dome C. The pure climatic signal has been resampled at a fixed temporal resolution of 5.8 a, to match the sampling rate of the top of the ice core (light red), and of 100 a (dark red). (a) Intermittent virtual core (light green: ice core resolution, dark green: 100 a averages) and running correlation between the intermittent virtual core and the climatic signal for different block-averaging windows. (b) Intermittent and diffused virtual core (light blue: ice core resolution, dark blue: 100 a averages) and running correlation between the intermittent and diffused virtual core and the climatic signal for different block-averaging windows.

As a visual representation of our findings, we present a simplified calculation of the impact of precipitation intermittency and diffusion on a long-term temperature time series (TRACE21k; Liu et al., 2009) for Dome C (Fig. 8; see Supplementary Materials S6). Precipitation intermittency adds a large amount of non-climatic noise, clearly visible at the resolution applied to extract data from the Dome C ice core (Fig. 8a, light green curve). While the standard deviation for the intermittent virtual core

for the last 1000 years is $\sigma_{1\,ka} = 2.6\,K$, the climatic signal has a standard deviation of only $\sigma_{1\,ka} = 0.64\,K$. The intermittent virtual core and the climatic signal are uncorrelated during stadial periods (Holocene and Last Glacial Maximum), and only show a notable correlation during the climate transition after averaging to time scales larger than 60 years ($r^2 \geq 0.5$). This is

expected as the scaling of Trace21k is only $\beta = 0.27$, much smaller than the values expected from ice core records (around 0.6; Münch and Laepple, 2018) and thus associated with a poor SNR according to our results.

The diffused virtual core sampled at the ice core resolution has a variance ($\sigma_{1\,ka} = 1.5\,K$) that is lower than for the intermittent virtual core but still larger than for the climatic signal (Fig. 8b). Although we only included the present level of precipitation intermittency and isotopic diffusion, this value is of the same order of magnitude as the one obtained for the actual temperature reconstruction from the Dome C ice core ($\sigma_{1\,ka} = 0.94\,K$; EPICA, 2004).

Overall, our results are in agreement with the call for caution made by Sime et al. (2009) when interpreting isotopic composition fluctuations in individual ice core records. We have shown here that across Antarctica, precipitation intermittency adds a significant noise component to the water isotope signal in ice cores due to the aliasing of the seasonal cycle (Persson et al., 2011). Using spectral methods, we could determine the lower limit for the time scales at which the ice core signal is sufficiently correlated with the climatic signal (Table 3). Isotopic composition profiles from snow pits on the East Antarctic Plateau exhibit a systematic visual similarity apparent to cycles with a period of roughly 20 cm (Casado et al., 2018) mostly due to diffusion of a signal dominated by white noise (Laepple et al., 2018). Our results indicate that a large part of the noise that needs to be added to the climatic signal is due to precipitation intermittency (63% of the initial variance on average across all of Antarctica).

## 4.2 Additional impact of stratigraphic noise

All three snow pit sites that we compared to our model outputs point toward additional noise that needs to be added prior to diffusion. Stratigraphic noise could be a likely candidate for this missing noise (Fisher et al., 1985), which is mostly white (Münch and Laepple, 2018) and results from a range of processes that affect the snow while it remains at the surface such as wind blowing (Groot Zwaaftink et al., 2013), sublimation and condensation (Casado et al., 2016; Ritter et al., 2016; Genthon et al., 2017), and surface metamorphism (Picard et al., 2012; Casado et al., 2018). Stratigraphic noise would further reduce the correlation between the climatic signal and the ice cores. In particular, it decreases the relative amount of variance associated with the seasonal cycle.

The amount of additional noise needed for the model outputs to match the PSDs of the observations matches the amount of stratigraphic noise obtained independently in Antarctica. Using the correlation between two trenches at EDML, Münch et al. (2017) estimated that the stratigraphic noise for the site of EDML accounts for 50% of the total signal, which is of the same order of magnitude as our estimate of the additional noise of 60% (Fig. 7). No corresponding estimate exists for Dome C. We do know, however, that the amount of snow accumulating at Dome C corresponds to only 10% of the amount of snow being deposited (i.e., about 90% is blown by wind several times before settling definitively) (Picard et al., 2019), which would suggest a similar amount of stratigraphic noise as our estimate of 80 % of the total variance.

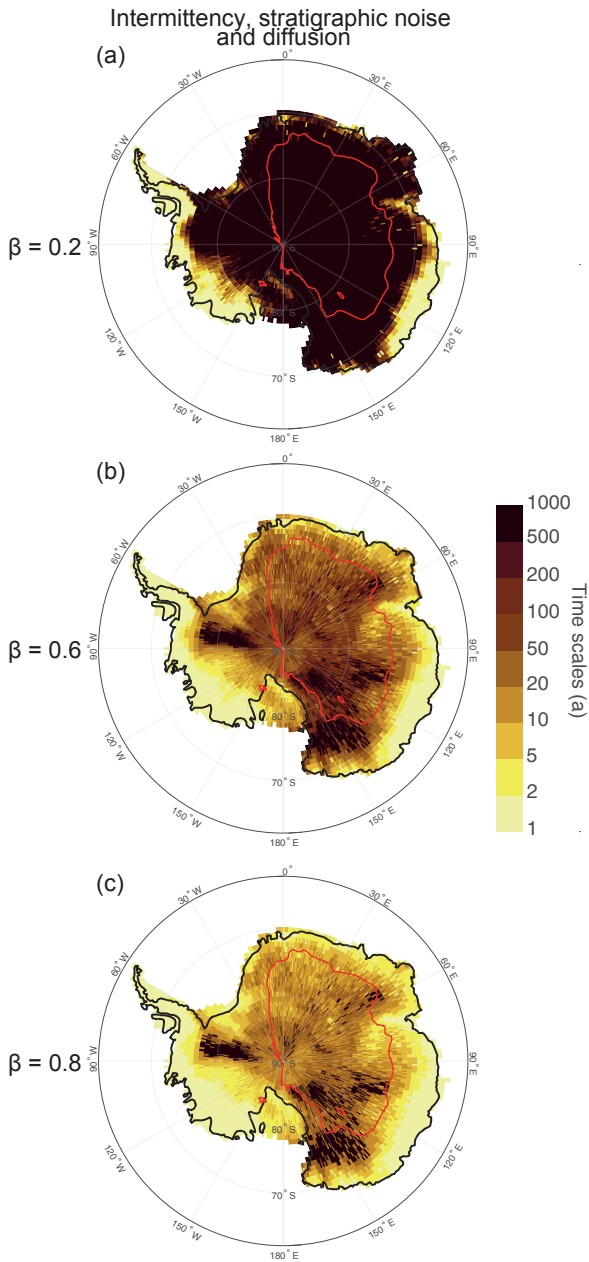

**Figure 9.** Maps of the signal retrieval time scales $\tau_a$ after block-averaging including the effects of precipitation intermittency, stratigraphic noise corresponding to 60% of the climatic signal's variance, diffusion, and additional measurement noise of 0.1 ‰$^2$, assuming climate spectrum scaling exponents of (a) $\beta = 0.2$, (b) $\beta = 0.6$ and (c) $\beta = 0.8$. The red line marks the contour of an accumulation of 25 cm w.e., see also Fig. S1.

For a stratigraphic noise level of 60% of the total variance of the climatic signal, we obtain on average a doubling of the values for the signal retrieval time scales $\tau_a$ (Fig. 9). We expect stratigraphic noise to have a different spatial pattern than the noise associated with precipitation intermittency, as both processes involve different physical mechanisms: wind blowing, sub-limation and condensation, metamorphism in case of stratigraphic noise versus precipitation formation in case of precipitation intermittency. An additional quantitative evaluation of the amount of stratigraphic noise with respect to the total variance of isotopic records would be necessary to be able to parameterise stratigraphic noise in our forward model.

While both stratigraphic noise and precipitation intermittency add white noise to the climatic signal, it is important to distin-guish the spatial and temporal properties of the white noise in each case. Stratigraphic noise from two locations separated by only a few metres will be essentially uncorrelated (Münch et al., 2016), while precipitation intermittency can exhibit a correla-tion across areas as large as $100 \times 100 \, \mathrm{km}$, which is suggested by an analysis of the spatial extent of the precipitation simulated by a regional climate model (Agosta et al., 2019). As a result, any attempt to increase the SNR by averaging several ice cores will need to take into account the different decorrelation lengths of both these noise sources (Münch and Laepple, 2018). On the one hand, to reduce the impact of stratigraphic noise, it would be sufficient to average two or more ice cores collected 10 m apart from a single site. On the other hand, to reduce the impact of precipitation intermittency, it will be necessary to collect the two ice cores from further appart. This, however, may introduce a bias, since the two sites which are further apart may have slightly different local temperature variations, and further biases may be introduced by dating uncertainties.

As a result, if one was able to make a large number of cores at a single site, the limiting signal retrieval time scale would be the one without considering the impact of stratigraphic noise, as presented in Fig. 6. In the typical case of having just one core available for a given site, the signal retrieval time scale is the one that includes stratigraphic noise (Fig. 9).

### 4.3 Limits of the present methodological approach

The first approximation on which we based the calculation of the signal retrieval time scales is that the amount of white noise generated by both precipitation intermittency and stratigraphic noise has remained constant through time. This approximation may not hold as precipitation patterns and amounts may have been different in the past, and thus change our postulated time scale limits. General Climate Models (GCM) can provide estimations of the changes in precipitation intermittency in the past, especially if they are linked to changes in the atmospheric circulation.

Furthermore, we needed to make certain assumptions about the spectrum of the climatic signal. Using the CMIP5 climate model simulations to estimate the spectrum of the climatic signal yields signal retrieval time scales $> 1000 \, \mathrm{a}$ for ice cores in Central Antarctica. These results stand in contrast to *in-situ* observations (Münch and Laepple, 2018), which suggest that the true regional climatic variability may be higher than predicted by GCMs (Laepple and Huybers, 2014), leading to more optimistic results.

This study provides only a lower boundary of signal retrieval time scales while taking into account what we believe to be the major contributions. However, there are several additional processes that affect the isotopic signal. First, dating affects the quality of the retrieved signal. Here, we illustrate a realistic case where dating interval between tie points is roughly 40 years (constrained by the length of the ERA-interim time series). We also used a perfect dating by tagging each layer of snow with a date and evaluate the impact on the presented results (Supplementary Material S3). For a real ice core, the uncertainty associated with the dating, as well as the variable accumulation amounts in between tie points also impacts the signal, leading to additional effects than the one described here (Supplementary Material S3).

Second, we did not include the effect of clear sky precipitation, which, in the center of Antarctica, can sum up to a significant fraction of the annual accumulation (Fujita and Abe, 2006) and tends to occur more evenly over the year, in contrast with the intermittent aspect of long-range precipitation (the only one included accurately in ERA-Interim). Such a more regular input of the signal would reduce the amount of noise created by precipitation intermittency. Sensitivity tests show that if half of the precipitation originates from clear sky precipitation, the signal retrieval time scales after precipitation intermittency are significantly larger (3.4 years instead of 17 at EDML for $\beta = 0.6$; see Supplementary Material S4). However, as the relationship between clear sky precipitation isotopic composition and temperature likely differs from the one between long range precipitation and temperature (Dittmann et al., 2016; Stenni et al., 2016), we expect additional noise from mixing these two different signal sources which would decrease again the signal retrieval time scale (3.9 years; see Supplementary Material S4). These limitations are overall linked to uncertainties in the surface mass balance (Genthon et al., 2015; Picard et al., 2019), which in Central Antarctica are quite large, in particular due to analytical limits to evaluate the precipitation amounts (both clear sky and long range), the snow blown and redeposited by wind (Palm et al., 2017), and sublimation and condensation (Genthon et al., 2017). In Sect. 4.2 we showed that the total noise level appears to be higher than predicted by just precipitation intermittency alone, which would suggest that the reduced white noise level generated by precipitation intermittency when including clear sky precipitation would have to be compensated by additional noise needed after precipitation intermittency (provided that we know the total noise level).

Third, exchanges between the surface snow and the atmosphere can lead to significant changes of the snow isotopic composition in between precipitation events in polar regions (Steen-Larsen et al., 2014; Ritter et al., 2016; Casado et al., 2018). These effects are not directly taken into account here, and they would take part after the precipitation has been deposited while the snow remains at the surface (Münch et al., 2017). On the one hand, snow–atmosphere exchanges would be more continuous over time and thus reduce the noise level in case these effects play a significant role compared to the isotopic variations driven by the isotopic content of the snowfall. On the other hand, sublimation and condensation could result in an addition of noise at the surface due to the strong spatial (linked to dune and sastrugi) and temporal (linked to variable cloud cover) variability in latent heat created mostly by radiative processes (Vignon et al., 2017).

For taking isotopic diffusion into account, we assumed it to be constant by using a diffusion length that corresponds to a depth of $100\,\mathrm{m}$, comparable to the lock-in depth. In reality, the amount of diffusion varies with depth as a result of firn diffusion (Johnsen, 1977; Johnsen et al., 2000; Laepple et al., 2018), layer thinning and ice diffusion (Pol et al., 2014). This can easily be included in our approach through the use of a more complete transfer function of diffusion which requires prior knowledge of the variations of the diffusion processes.

    Our present approach is not suitable to actually produce realistic depth series of isotopic composition of snow in Antarctica, mainly because of the uncertainties of the ERA-Interim precipitation time series and the difference between snow accumulation and precipitation in Antarctica due to wind blowing, sublimation and condensation, and other processes involved in the surface mass balance. For instance, while it is well known that ERA-Interim predicts rather well the precipitation timing but
shows biases for the magnitudes (Medley et al., 2013), we expect that the produced virtual cores from our model suffer random dephasing due to the errors on the amount of accumulation linked with every event of precipitation. In addition, Picard et al. (2019) showed that the snow for a specific point does not actually accumulate necessarily during the precipitation event, which would create additional random lags between our produced virtual cores and actual snow pits. This does not affect the present results which are based on spectral analysis and thus independent of the phase.

    Finally, another important aspect that limits the ability of our modelling approach to provide minimum time scales at which an actual ice core can be used is the lack of constraints on the strength of decadal to centennial climate variability in Antarctica, reflected here by the values of $\beta$, both in general as well as at the specific ice core sites. We were not able to estimate the scaling for each site and instead presented exemplary results for the values of $\beta$ of 0.2, 0.6 and 0.8 to cover the whole range of
reasonable assumptions for the climate variability of the last millennia. The lowest value of $\beta$ of 0.2 represents the variability as simulated from current climate models but is likely pessimistic as current climate models tend to underestimate regional climate variability (Laepple and Huybers, 2014). It further reflects the small scaling of climate variability that was found in firn-cores of the WAIS region (Münch and Laepple, 2018). The value of $\beta = 0.6$ represents our best guess based on a single study that estimated the decadal-to-centennial scaling exponents from an array of ice core records in the EDML region (Münch
and Laepple, 2018). Finally, the highest value of $\beta$ of 0.8 represents an optimistic assumption of strong slow climate variability, similar to estimates for regional ocean variability of $\beta = 1$ (Laepple and Huybers, 2014). Our choice is based on the reasoning that while the climate variability scales less on land than over the ocean on interannual to decadal time scales (Huybers and Curry, 2006), both should converge to a similar scaling behaviour on longer time scales (North et al., 2011). Further analyses of firn and ice-core arrays, paired with a better understanding of the noise processes such as presented here, would provide
means to make significant progress in the interpretation of ice core records and Antarctic climate variability.

## 5 Conclusions

We provided a forward modelling approach to estimate the minimum time scales at which meaningful (SNR $\geq 1$) signals can be extracted from ice cores, taking into account the potential effects from (i) precipitation intermittency, (ii) diffusion, and (iii) measurement noise. This was achieved by estimating the spectral properties of these three processes using ERA-Interim time series of temperature and precipitation.

Our results underline that the ability to reconstruct past climate conditions from ice cores depends not only on the noise levels imposed by precipitation intermittency and stratigraphic noise, but also on the strength of the input signal. As a result, a particularly strong signal, such as the deglaciation, will be imprinted in the ice cores at a much higher effective resolution than the limited Holocene temperature variations in Antarctica. Potential variations in the noise levels during past climatic conditions will also strongly affect our results.

The systematic analysis of the various processes that affect how climatic signals are stored is important for high-resolution climate reconstructions. We propose that the use of spectral properties, rather than linear correlations in a calibration period of the proxy with instrumental observations, provides a great potential to quantitatively estimate the signal recorded in the isotopic composition in ice cores.

*Code availability.* The code is available on Mathworks: https://fr.mathworks.com/matlabcentral/fileexchange/77095-virtual-firn-core-generator?s_tid=prof_contriblnk.

*Author contributions.* MC and TL designed the research. MC performed the analysis and wrote the first draft of the paper. MC, TM, and TL contributed to the interpretation and to the preparation of the final paper.

*Competing interests.* The authors declare no competing interests.

*Acknowledgements.* The research leading to these results has received funding from the Alexander von Humboldt Foundation (Germany), project DEAPICE. This project was further supported by Helmholtz funding through the Polar Regions and Coasts in the Changing Earth System (PACES) programme of the Alfred Wegener Institute, by the Initiative and Networking Fund of the Helmholtz Association Grant VG-NH900 and by the European Research Council (ERC) under the European Union's Horizon 2020 research and innovation programme (grant agreement no. 716092). The work profited from discussions at the CVAS working group of the Past Global Changes (PAGES) programme.

We acknowledge the help of Igor Kröner, Raphael Hebert, Amaelle Landais, Jilda Alicia Caccavo, Tyler Jones and Geoffrey Gourdet, and our fruitful discussions. We acknowledge Sentia Goursaud and one anonymous reviewer for their fruitful comments.

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
