# Peer review of "Climatic information archived in ice cores: impact of intermittency and diffusion on the recorded isotopic signal in Antarctica"

_Climate of the Past, 2019_

## Referee Comment (RC1) · Sentia Goursaud (Referee) · 23 Dec 2019

Referee : Sentia Goursaud

Summary This paper models a step by step water stable isotope ice core data distinguishing the climatic signal from intermittent precipitation and diffusion noises, using ERA-interim reanalyses. The analyses are then based on spectral observations. The identification of specific frequencies for a signal-to-noise equal to one allows a quantification of the resolution to consider to extract a climatic signal. These kinds of results are undoubtely very needed, and the approch proposed in this study could be very useful for ice core people. However, in this current state, the paper needs some read-

justments to make it easier to understand as well as more applied to specific sites, as outlined in the general comments. I thus highly recommend the publication of this paper, but after major modifications.

General comments - No dedicated paragraph can be found for the data. Could you introduce the ERA-interim reanalyses, CMIP5 models and snow pit data in a first paragraph of section 2 ? Also, no justification was given neither for ERA-interim : why not ERA5 wich show overall better performances ? See Vignon, Étienne, Olivier Traullé, and Alexis Berne. "On the fine vertical structure of the low troposphere over the coastal margins of East Antarctica." Atmospheric Chemistry and Physics 19.7 (2019): 4659-4683. Could you justify the choice of the 5 models among the CMIP5 models ? Finally, please give more details and references for the snow pit data you used.

- As you compare the outputs from your virtual cores, the precipitation data used should be the more realistic possible. Your method using ice core data from Thomas et al. is of great subtilty, as the SMB data from Vaughan et al. was not checked by Athern et al. before their interpolation. However, could you make a prior validation of your method, by comparing it against data, for instance from Favier et al. : Favier, V., Agosta, C., Parouty, S., Durand, G., Delaygue, G., Gallée, H., ... & Krinner, G. (2013). An updated and quality controlled surface mass balance dataset for Antarctica. Or using the SMHiL data : Agosta, C., Favier, V., Krinner, G., Gallée, H., Fettweis, X., & Genthon, C. (2013). High-resolution modelling of the Antarctic surface mass balance, application for the twentieth, twenty first and twenty second centuries. Climate dynamics, 41(11-12), 3247-3260. Once robustly demonstrated, this could be reused.

- Some points of your method require clarity, as an explanation on the law power, and better explanations on the difference between $\tau$a and $\tau$b.

- The paper seems to be directed to ice core users, thus for applied purposes. I would suggest to better introduce which time scales analyses might be affected by the precipitation intermittency and the diffusion. For instance, the glacial-interglacial variability is

so strong, that it is not pointed in the line of sight of such questions. So please frame the use of your analysis for Holocene reconstructions. Also, it would be very practical to get quantitative frequency thresholds ($\tau$, associated with the best $\beta$ fit) for every site you cited in Table 4, similarly as your work for EDML in Figure 4. That would be of a high contribution.

- Finally, could you specify for each correlation if it is significant (p<0.05) ?

Specific comments p1 l1-l3 : The sentence is confusing as one might think water stable ice core records result only from the surface temperature, the intermettency of precipitation and firn diffusion. p1 l8 : You are actually not giving a proper transfer function in the paper, that we could apply to extract the climatic from the whole record.

p1 l22 : please add Âń of the climatic signal Âż after Âń the temporal resolution Âż.

p1 l23 : please add more recent references for coastal sites, e.g. : Caiazzo, L., Baccolo, G., Barbante, C., Becagli, S., Bertò, M., Ciardini, V., ... & Gabrieli, J. (2017). Prominent features in isotopic, chemical and dust stratigraphies from coastal East Antarctic ice sheet (Eastern Wilkes Land). Chemosphere, 176, 273-287. Goursaud, S., Masson-Delmotte, V., Favier, V., Preunkert, S., Legrand, M., Minster, B., & Werner, M. (2019). Challenges associated with the climatic interpretation of water stable isotope records from a highly resolved firn core from Adélie Land, coastal Antarctica. The Cryosphere, 13(4), 1297-1324. Vega, C. P., Schlosser, E., Divine, D. V., Kohler, J., Martma, T., Eichler, A., ... & Isaksson, E. (2016). Surface mass balance and water stable isotopes derived from firn cores on three ice rises, Fimbul Ice Shelf, Antarctica. The Cryosphere, 10(6), 2763-2777.

p1 l8 : change Âń however Âż to Âń moreover Âż>

p2 l28 : For low accumulation sites, I would recommend citing : Frezzotti, M., Urbini, S., Proposito, M., Scarchilli, C., & Gandolfi, S. (2007). Spatial and temporal variability of surface mass balance near Talos Dome, East Antarctica. Journal of Geophysical

Research: Earth Surface, 112(F2).

p3 l14 : Remove Âń while Âż before Âń ...the actual value of Âż, and change Âń scale Âż by Âń are scaled Âż. I do not understand the end of your sentence related to uncertainty. Can you explain ?

p3 l21-22 : Âń This yields an intermittent virtual core ... Âż Please rewrite the sentence to make it understandable.

p4 l21 : change Âń providing Âż with Âń provided with Âż.

p4 l23 : Please give more details. Is it acceptable within the range of variability ? Which is ? Also, you gave references mostly related to the Plateau. However, the outputs migth be optimistic when accounting for coastal areas and the Peninsula, see Figure 2 from : Goursaud, S., Masson-Delmotte, V., Favier, V., Orsi, A., & Werner, M. (2018). Water stable isotope spatio-temporal variability in Antarctica in 1960–2013: observations and simulations from the ECHAM5-wiso atmospheric general circulation model. Climate of the Past, 14(6), 923-946.

p5 l5 : change Âń had been themselves corrected Âż by Âń were preliminary corrected Âż.

p5 l25 : There might be much more recent publications related to the random occurency of precipitations. Please update the litterature. I would recommend citing : Turner, John, et al. "The Dominant Role of Extreme Precipitation Events in Antarctic Snowfall Variability." Geophysical Research Letters 46.6 (2019): 3502-3511.

p6 l12 : remove Âń for Âż before Âń the SNR for a given frequency Âż.

p6 l11 : - For people not familiar with the tools you usem this part of the paper is not easy to understand. Especially, it is difficult to follow the logic of the following paragraphs. What could help is to first give the role of each tool you use, ie SNR, $\tau$a and $\tau$b prior to the way they are obtained. - Âń either in the sampling, or by average samples Âż : I do not understand.

p9 l5 : move Âń only Âż after Âń conditions Âż.

p9 l8 : Could you define the Âń amount of lost variance Âż ? Is it the difference of variance between the climatic virtual core and the intermittency virtual core ? Or the variance of a core which is reduced with increasing frequencies ?

p9 l9 : please specify that the warm bias comes from ERA, and that it is emphasized on the plateau, thus referring to a more comprehensive study as : Fréville, H., Brun, E., Picard, G., Tatarinova, N., Arnaud, L., Lanconelli, C., ... & Van den Broeke, M. (2014). Using MODIS land surface temperatures and the Crocus snow model to understand the warm bias of ERA-Interim reanalyses at the surface in Antarctica. The Cryosphere, 8(4), 1361-1373. If you rather report to the higher amount of precipitation during summer compared to winter, you should not speak about bias. Please make it clear. But in all cases consider my previous comment, at least of the ERA description.

p9 l11 : Is this correlation significant ?

p9 l12 : r2=0.34 is not that high, so I would rather suggest a part explanation of lost variance due to precipitation intermittency.

p9 l13 : The sentence Âń Nevertheless... Âż is confusing because it does not refer to the correlation coefficient, but I guess it does ? If so, could you thus gather the two sentences the sentences related to the mean and range values of this correlation coefficient or give more clarity in anyway ?

p9 l18 : Here I do not understand what you used to compute the correlation coefficient, once more probably because It is not clear to me what the amount of lost variance is.

p9 l22 : Âń r=0.22 Âż. Previously, you gave $r^2$ and not r. Could you check it was your intention to give r here. Is it significant ? What is r between the diffused virtual core and the intermittent virtual core ?

p9 l28 Âń the seasonal cycle clearly dominates the signal by roughly two orders of magnitude Âż, compared to another time scale ? Please specify.

p10 l11 : remove the dot before Âń see section Âż.

p11 l6 : Even if the PSD dislpays the square of the amplitude, could you change it to the amplitude (thus in ‰, so it is more relevant for ice core people.

P12 l6 : Can you give more information on the power law for the PSD, especially the way it is been found. You refer to an entire book, so it is not easy to find the proper dedicated paragraph.

p13 l1-5 : Please add a robust justification for illustrating the power law only for $\beta$=0.6 and $\beta$=0.8.

p13 l5-8 and p15 l5-7: To support your argument, it would worth to add contours of accumulation in Fig. 5 and to comment it there.

p15 l8 : which snow pit data were used ? Could it be introduced into the data section ?

p17 l2 : change Âń bellow Âż to Âń below Âż.

p17 l6 : change Âń an Âż to Âń a Âż.

p19 l3 add Âń as Âż to Âń such Âż.

Figures Figure 2 : That is a very nice plot !!!

Figure 3 : Are all simulated relationships signficant ? Please remove are for not significant relationships, are put hatches. Could you add contours for r from r=0.9 and 0.8, to point areas where the intermittency does relatively weakly affect the climatic signal ? Could you use the same colours for intermittency only, and intermittency and diffusion plots to make the comparision easier.

Figure 4 : could you add a vertical line for the seasonal frequency so it could illustrate better lines 6-7 in the text above the figure. Also detail in the legend that these fits and outputs correspond the the EDML site.

Figure 5 : see the specific comment suggesting adding coutours of accumulation.

Figure 6 and 7 : please complete the description presently suspended Âń have affected the signal... Âż.

Tables Table 4 : I do not see where table 4 has been cited in the text. It shows that the results are very $\beta$-dependant, and that it would have been expected similar fits than those displayed on Figure 4, in order to point the $\beta$ value to consider, and associated $\tau$ for each site.

---

## Referee Comment (RC2) · Anonymous Referee #2 · 18 Jan 2020

Review of Casado et al, Climatic information archived in ice cores. . .

Casado et al. develop a model to understand the timescale on which traditional water isotopes (dD and d18O) reflect climate. They assess two processes: diffusion and intermittency of precipitation. The primary result is a lower bound for the timescale at which climate variability can be reconstructed, which ranges from a year on the coast to a millennium in the interior. The manuscript is well written and the topic is of interest as the ability to measure water isotopes at cm resolution is becoming common. The underlying modeling is sound with a strong emphasis of spectral techniques. The results of what timescale of climatic information ice cores will contain are somewhat

underwhelming, since they vary by two orders of magnitude based on the assumed exponent for the power law representing long timescale climate variability. However, the primary interest of this work is not the conclusion, but rather the method development for a more quantitative understanding of processes limiting climate interpretation.

The assumption of the power law variability and the appropriate exponent does feel neglected in the paper. The authors use 1000-year runs of GCMs to estimate the climate variability, then seem to not like the result, so basically punt on the issue and use a range from 0 to 1, although the discussion primarily focuses on 0.6 (from a single study at EDML but seems to ignore the differences with WAIS in that same study) and 0.8. Someone reading this paper quickly would think that the appropriate range would be 0.6 to 0.8 because in three different figures, these are the only values plotted. A much fuller discussion of the appropriate value needs to be included so that the reader can understand not just the impact, but the state of knowledge. A single reference to one of the group's previous papers in CP is not enough for an assumption that dominates the results.

The forward model also seems to neglect two important processes: clear-sky precipitation and atmosphere-surface snow isotopic exchange. Both of these have the potential to offset the effect of precipitation intermittency. Assuming ERAi captures the timing of precipitation is fine in high precipitation areas, but these are also of the least interest since they receive precipitation relatively consistently. This assumption is much less valid for central East Antarctica, where clear-sky precipitation can make up a significant portion of total precipitation due to the infrequency and low volume of precipitation events. As to atmosphere-surface snow exchange, quantitative modeling would be challenging given the relative lack of information on the topic, but a qualitative discussion of its effect would be valuable.

Overall, this paper represents useful work that with a few improvements will contribute to water isotope analysis in ice cores.

Specific comments: This manuscript does not have line numbers, which should be remedied for future CP discussion papers because it makes referring to specific instances very difficult for both the referee and the authors.

- physico-chemical > why create this abomination of a hyphenation? Just write it out

- P2, last paragraph, consider using duration instead of size qualifiers, so replace larger with longer when referring to a timescale

- P2, last paragraph, change "read" to "interpret"

- P2, last paragraph, I think "virtually unlimited" is an overstatement given dispersion in CFA systems is still a limiting factor in the effective resolution

- P4, third paragraph, I'm really confused about what is being said in this paraph about perfect dating. I think I follow that because you tag the age of layers in the virtual core, you can compare these without age uncertainty to the original climate signal. However, you don't explain how you then stretch between the tagged ages. And the discussion of the depth scale getting out of phase seems both obvious and also not relevant, adding confusion.

- P4, last paragraph. This paragraph misrepresents the impacts of wind scour. The references (i.e. Picard et al., 2019) appear to be focused solely on interior East Antarctica, whereas the authors here imply that Antarctica as a whole can lose 90% of its accumulation to wind scour, thus explaining the ERAi overestimation of total precipitation. The authors need to be specific about when they are writing about Antarctica as a whole, and when only specific regions. ERAi is well known to get precipitation timing correct, but magnitudes off, with Medley et al. 2013, GRL being a good reference.

- P5, first full paragraph, how can the satellite data of Arthern 2006 have been corrected to the data in a 2017 paper. Be clear with the reference. Either cite older sources, or be specific about what data in the Thomas paper was used.

- P6 – define SNR in an equation. It gets confusing because you use SNR in an

[Figure]

equation before your present an alternate SNR(f) in a subsequent equation.

- Figure 2: This is a well crafted figure that illustrates the methods well

- Figure 4: This figure is hard to interpret. There seems to be about every shade and thickness of red line possible. Differentiate the climate model coloring. Describe the beta=0.2 fit. Label each of the power law exponent lines on the left, rather than just varying the thickness. Or maybe you want to rethink this figure all together and break it into parts, more like Figure 2.

- Combine Tables 1 and 2 and use the caption as an opportunity to distinguish what tao_a and tao_b are.

- Figure 5, the use of just 0.6 and 0.8 feels misleading given the uncertainty in this parameter choice. At the least, I think beta=0.2 needs to be included, since it will show up in table 4. Same goes for Figures 6 and 9. Also, all these figures should have the ice core locations of sites in Table 4 shown.

- Table 3: I'm not sure what tao without a subscript is. Is this tao_a like in the caption?

- P17, "below" not "bellow", but I would like to see "bellow" used in a CP paper sometime

- P20: section 4.3, I appreciate the inclusion of a section like this, but it seems like many limits of the approach were not discussed. Expand this with discussion of beta and atmosphere-surface snow isotopic exchange.

---

## Author Comment (AC1) · 6 Apr 2020

We would like to thank the two reviewers for their comments. We have been working toward a new version of the manuscript taking their respective comments into account. We propose here responses to their comments with examples of the changes we envision taking. We include the comments from the reviewers in black, our responses in blue, and the modifications to the manuscript in red in this response file.

**Referee #1: Sentia Goursaud**

Summary

This paper models a step by step water stable isotope ice core data distinguishing the climatic signal from intermittent precipitation and diffusion noises, using ERA-interim reanalyses. The analyses are then based on spectral observations. The identification of specific frequencies for a signal-to-noise equal to one allows a quantification of the resolution to consider to extract a climatic signal. These kinds of results are undoubtely very needed, and the approch proposed in this study could be very useful for ice core people. However, in this current state, the paper needs some readjustments to make it easier to understand as well as more applied to specific sites,as outlined in the general comments. I thus highly recommend the publication of thispaper, but after major modifications.

General comments

- No dedicated paragraph can be found for the data. Could you introduce the ERA-interim reanalyses, CMIP5 models and snow pit data in a first paragraph of section 2 ? Also, no justification was given neither for ERA-interim : why not ERA5 wich show overall better performances ? See Vignon, Étienne, Olivier Traullé, and Alexis Berne. "On the fine vertical structure of the low troposphere over the coastal margins of East Antarctica." Atmospheric Chemistry and Physics 19.7 (2019): 4659- 4683. Could you justify the choice of the 5 models among the CMIP5 models ? Finally, please give more details and references for the snow pit data you used.

> The section 2.2 ("Input time series and correction") was used to describe ERA-interim and the corrections made on it. In this same section, we included more details on the 8 (not 5) CMIP5 models that are used here. As we needed to have 1000-year long time series to produce the spectral estimates on longer time scales, we used the past1000 runs. Here, we used all of the 8 past1000 runs which were available at the time of the data processing. A description of these models as well as a link to the article presenting them has been included in the revised version of the manuscript (Line 146 to 150):

> "In addition, we use the millennial CMIP5 climate model simulations to compare our results with longer time series than the ones produced with ERA-Interim. We use the past1000 simulations from eight General Climate Models (GCM), namely BCC-CSM1-1, CCSM4, CSIRO-Mk3L-1-2, FGOALS-gl, GISS-E2-R, IPSL-CM5A-LR, MIROC-ESM, MRI-CGCM3 that cover the last 1000 years and include the historical solar and volcanic forcing \citep{Bothe2013}"

> We choose ERA-interim as ERA5 was not available on a period as long as ERA-interim when we prepared this manuscript. While the study of Vignon et al. shows that ERA5 has better performances than ERA-interim, we do not expect a significant impact of the use of a different reanalysis on the results. Indeed, the sensitivity test shown in the supplementary materials on the impact of the correction of the precipitation amount suggests that the precision of the input precipitation data is not very important, as long as the precipitation data is largely intermittent.

> We include more details on the snow pit data we used, in particular including the different studies in which they were used before. This is part of a new sub-section in section 2, distinct from section 2.2 in order not to mix input to the model, and other datasets we are comparing our results with.

- As you compare the outputs from your virtual cores, the precipitation data used should be the more realistic possible. Your method using ice core data from Thomas et al. is of great subtility, as the SMB data from Vaughan et al. was not checked by Athern et al. before their interpolation. However, could you make a prior validation of your method, by comparing it against data, for instance from Favier et al. : Favier, V., Agosta, C.,

Parouty, S., Durand, G., Delaygue, G., Gallée, H., ... & Krinner, G. (2013). An updated and quality controlled surface mass balance dataset for Antarctica. Or using the SMHiL data : Agosta, C., Favier, V., Krinner, G., Gallée, H., Fettweis, X., & Genthon, C. (2013). High-resolution modelling of the Antarctic surface mass balance, application for the twentieth, twenty first and twenty second centuries. Climate dynamics, 41(11- 12), 3247-3260. Once robustly demonstrated, this could be reused.

> We agree with the reviewer that robust estimates of the precipitation amount in large parts of Antarctica, especially in the centre are missing. However, our results are largely insensitive to the absolute amount of precipitation as the determining factor, the ratio of the climate signal vs. the noise from intermittency is not affected. We demonstrate this as an example in the supplementary information. Including diffusion, our results show some sensitivity on the accumulation amount but even this effect is small and this doesn't affect our main conclusions. In addition, using a spectral approach, it is not important to have the exact date of precipitation events as long as we obtain a correct distribution of precipitation events, one of the hypotheses of our approach which is discussed in section 4. Nonetheless, we did not discuss that if one wanted to make a comparison of the modelled snow pits with actual snow pits from a specific site, having the most accurate precipitation estimates would then be very relevant. We will include a discussion about this (lines 478 to 486):

> "Our present approach is not suitable to actually produce realistic depth series of isotopic composition of snow in Antarctica, mainly because of the uncertainties of the ERA-interim precipitation time series and the difference between snow accumulation and precipitation in Antarctica due to wind blowing, sublimation and condensation, and other processes involved in the surface mass balance. For instance, while it is well known that ERA-interim predicts rather well the precipitation timing but shows biases for the magnitudes \citep{Medley2013}, we expect that the produced virtual cores from our model suffer random dephasing due to the errors on the amount of accumulation linked with every event of precipitation. In addition, \cite{Picard2019} showed that the snow for a specific point does not actually accumulate necessarily during the precipitation events, which would create additional random lags between our produced virtual cores and actual snow pits. This does not affect the present results which are based on spectral analysis and thus independent from the phasing."

- Some points of your method require clarity, as an explanation on the law power, and better explanations on the difference between τ a and τ b.

> See also the specific comments. We included more explanation for the difference between $\tau_a$ and $\tau_b$ as well as additional references explaining our motivation to assume a power law behaviour of the spectral density.

- The paper seems to be directed to ice core users, thus for applied purposes. I would suggest to better introduce which time scales analyses might be affected by the precipitation intermittency and the diffusion. For instance, the glacial-interglacial variability is so strong, that it is not pointed in the line of sight of such questions. So please frame the use of your analysis for Holocene reconstructions. Also, it would be very practical to get quantative frequency thresholds (τ , associated with the best β fit) for every site you cited in Table 4, similarly as your work for EDML in Figure 4. That would be of a high contribution.

> Indeed, as the reviewer mentions, we are only focusing on decadal to millennial variability, as detailed in section 4. We will add to the introduction of the revised version of the manuscript that the study focuses on time scales limited to 1000 years.

> We would like to note that it can still be relevant for high resolution analyses outside of the Holocene, for instance for rapid events such as Dansgaard-Oeschger cycles in the last glacial period. There are two parameters to consider here: the strength of the signal versus the amount of noise generated by precipitation intermittency. In this regard, even during the last deglaciation, our results could theoretically be applied in the sense that the signal is stronger, and this changes the time scale at which we can retrieve a signal. We tried to illustrate this in Fig. 8. Then obviously, going deeper into the firn,

additional processes need to be considered, such as thinning and ice diffusion, which would add up to what we present here.

We also agree that having values of τ for the best fit of β values for each site for the Holocene would be a great added value. Unfortunately, the only firn-core based estimate of β we have at hand comes from EDML and WAIS. At these sites enough shallow ice cores were analysed to separate the climate variability from non-climate noise. As a result, we could only generalise the values of β observed at EDML for all the sites (0.6). In the revised version of the manuscript, we include maps for values of β of 0.2, in addition to 0.6 and 0.8, to cover a larger range of possible values of β. We also discuss in more detail the lack of determination of β for more sites in Antarctica, and why we have to use a range of values, instead of independent determination for each site (lines 499 to 511):

"We were not able to estimate the scaling for each site and instead presented results for values of $\beta$ of 0.2, 0.6 and 0.8 to cover the whole range of reasonable assumptions for the climate variability of the last millennia. The lowest value of $\beta$ of 0.2 represents the variability as simulated from current climate models but is likely pessimistic as current climate models tend to underestimate regional climate variability \citep{Laepple2014}. It further reflects the small scaling of climate variability that was found in firn-cores of the WAIS region (\citep{Munch2018}). The value of $\beta$ = 0.6 represents our best guess based on a single study that estimated the decadal-to-centennial scaling exponents from an array of ice core records in in the EDML region (\citep{Munch2018}). Finally, the highest value $\beta$ of 0.8 represents an optimistic assumption of strong slow climate variability, similar to estimates for regional ocean variability of $\beta = 1$ \citep{Laepple2014}. Our choice is based on the reasoning that while on interannual to decadal time-scales climate variability over land scales less than over the ocean \citep{Huybers2006}, they should converge to a similar scaling behaviour on longer-time scales \citep{North2011}. Further analyses of firn and ice-core arrays, paired with a better understanding of the noise processes such as presented here, would provide a mean to make significant progress in the interpretation of ice core records and Antarctic climate variability."

- Finally, could you specify for each correlation if it is significant (p<0.05) ?

We now specify the significance of correlation values in the text. For the maps, see our response below in the dedicated specific comment.

Specific comments

p1 l1-l3 : The sentence is confusing as one might think water stable ice core records result only from the surface temperature, the intermettency of precipitation and firn diffusion.

We fully agree with the reviewer that temperature, intermittency and firn diffusion are only parts of the processes leading to the isotope record. However, as we write "but also contains", we feel that our original sentence is correct

p1 l8 : You are actually not giving a proper transfer function in the paper, that we could apply to extract the climatic from the whole record.

We agree with the reviewer that we are not providing an analytical transfer function, but we do provide a transfer function in the sense that we describe the input of precipitation intermittency as a white noise level, and the diffusion as a low pass filter for which the equation is well demonstrated now. It is true that we have not provided the back transfer function to demodulate the noise and the diffusion out of the ice core and go backward from an ice core record to a climatic signal.

We modify this sentence to (lines 8 to 10):

"Here, we use reanalysis data (ERA-Interim) combined with satellite products of accumulation to evaluate the spatial distribution of the numerical estimates of the transfer function "

p1 l22 : please add " of the climatic signal " after " the temporal resolution ".

Modified accordingly.

p1 l23 : please add more recent references for coastal sites, e.g. : Caiazzo, L., Baccolo,G., Barbante, C., Becagli, S., Bertò, M., Ciardini, V., ... & Gabrieli, J. (2017). Prominent features in isotopic, chemical and dust stratigraphies from coastal East Antarctic ice sheet (Eastern Wilkes Land). Chemosphere, 176, 273-287. Goursaud, S., Masson- Delmotte, V., Favier, V., Preunkert, S., Legrand, M., Minster, B., & Werner, M. (2019). Challenges associated with the climatic interpretation of water stable isotope records from a highly resolved firn core from Adélie Land, coastal Antarctica. The Cryosphere, 13(4), 1297-1324. Vega, C. P., Schlosser, E., Divine, D. V., Kohler, J., Martma, T., Eichler, A., ... & Isaksson, E. (2016). Surface mass balance and water stable isotopes derived from firn cores on three ice rises, Fimbul Ice Shelf, Antarctica. The Cryosphere, 10(6), 2763-2777.

References included.

p1 l8 : change " however " to " moreover ">

Modified accordingly.

p2 l28 : For low accumulation sites, I would recommend citing : Frezzotti, M., Urbini, S., Proposito, M., Scarchilli, C., & Gandolfi, S. (2007). Spatial and temporal variability of surface mass balance near Talos Dome, East Antarctica. Journal of Geophysical Research: Earth Surface, 112(F2).

Reference included.

p3 l14 : Remove " while " before " ...the actual value of ", and change " scale " by " are scaled ". I do not understand the end of your sentence related to uncertainty. Can you explain ?

We include more details explaining why the change of slopes can affect the results due to the amplification of the measurement uncertainty (lines 77 to 81):

"A large range of slopes are reported in the literature, but in our case, the uncertainty on the value of the slope does not affect our main conclusions as both the input signal and the noise scale with the same coefficient. A different value of slope only affects our results when considering the impact of the measurement uncertainty. While the assumed isotopic signal and noise from intermittency scales with the slope, the measurement uncertainty stays constant and thus its relative impact is changing."

p3 l21-22 : " This yields an intermittent virtual core ... " Please rewrite the sentence to make it understandable.

We modify the sentence to:

"This yields an intermittent virtual core, whose total depth (in meters) is the product of the duration of the input signal (a) and the mean accumulation rate (m a$^{-1}$)."

p4 l21 : change " providing " with " provided with ".

We modify this sentence to:

"both from ERA-interim re-analysis at a temporal resolution of six hours"

p4 l23 :  Please give more details. Is it acceptable within the range of variability ? Which is ? Also, you gave references mostly related to the Plateau. However, the outputs migth be optimistic when accounting for coastal areas and the Peninsula, see Figure 2 from : Goursaud, S., Masson-Delmotte, V., Favier, V., Orsi, A., & Werner, M. (2018). Water stable isotope spatio-temporal variability in Antarctica in 1960–2013: observations and simulations from the ECHAM5-wiso atmospheric general circulation model. Climate of the Past, 14(6), 923-946.

We do not use ECHAM5-wiso, which in Goursaud et al, 2018 was nudged against ERA-40. However, it is true that the comparison we included were related to the Plateau conditions. We include a reference to Jones and Lister (2015) Int. J. Climatol, which covers more globally the comparison of in-situ measured temperature and ERA-interim in Antarctica.

p5 l5 : change Ân´ had been themselves corrected Âz˙ by Ân´ were preliminary corrected Âz˙.

Modified accordingly.

p5 l25 : There might be much more recent publications related to the random occurency of precipitations. Please update the litterature. I would recommend citing : Turner, John, et al. "The Dominant Role of Extreme Precipitation Events in Antarctic Snowfall Variability." Geophysical Research Letters 46.6 (2019): 3502-3511.

Reference included.

p6 l12 : remove Ân´ for Âz˙ before Ân´ the SNR for a given frequency Âz˙.

Modified accordingly.

p6 l11 : - For people not familiar with the tools you usem this part of the paper is not easy to understand. Especially, it is difficult to follow the logic of the following paragraphs. What could help is to first give the role of each tool you use, ie SNR, τ a and τ b prior to the way they are obtained. - Ân´ either in the sampling, or by average samples Âz˙ : I do not understand.

We followed the reviewer's suggestion and included a description of each tool before we start using them. Changes include (Lines 174 to 195):

"To evaluate the extent to which the climatic signal is preserved in the ice core record as a function of time scale, we assess the minimum time scales $\tau$ at which the SNR reaches a value of $1$. Since in any proxy record containing a climatic signal and noise, the SNR and the correlation between the record and the climatic signal are linked via:

\begin{equation}

r^2 = \frac{SNR}{1 + SNR}

\end{equation}

As a result, this minimum time scale will correspond to a correlation with the climate signal of $r = \sqrt{0.5}\sim0.71$.

\newline

The SNR can be defined in two ways. First, one can analyse the SNR at a specific frequency after filtering the ice-core time series with a narrow bandpass filter, in which case we refer to the minimum time scale as $tau_b$, where the subscript $b$ stands for bandpass. Second, and more commonly, the ice-core time series results from averaging a higher-resolution record to a fixed temporal resolution, either by discrete sampling in the depth domain or by averaging the dated record to a specific resolution, e.g. annual or decadal resolution. In this case, we refer to the minimum time scale as $tau_a$, where the subscript $a$ stands for averaging.

\newline

Formally, $\tau_b$ is given by the critical frequency $f_b=1/tau_b$, for which the direct ratio of the signal and noise spectra reaches $1$,

\begin{equation}

> SNR(f_b) =  \mathcal{S}(f_b)/\mathcal{N}(f_b) = 1
>
> \end{equation}
>
> where $\mathcal{S}$ and $\mathcal{N}$ are the PSDs of the signal and the noise. To obtain $tau_a$, the PSDs of the signal and noise have to be integrated to that critical frequency $f_a=1/tau_a$, for which the ratio of the \emph{integrated} spectra reaches $1$,
>
> \begin{equation}
>
> SNR(f) = \frac{\int_{1/L_{R}}^{f_a} \mathcal{S}(\nu) d\nu}{\int_{1/L_{R}}^{f_a} \mathcal{N}(\nu) d\nu} = 1
>
> \end{equation}
>
> where $L_R$ is the length of the record (either in years or in metres). Graphically, this is given by the ratio of the area representing the signal excess $\mathcal{A}_{signal}$ to the area representing the noise excess $\mathcal{A}_{noise}$ (Fig. 1a). If the signal is redder than the noise, the minimum time scale $tau_a$ will be smaller than $tau_b$.}"

p9 l5 : move Ân´ only Âz˙ after Ân´ conditions Âz˙.

> Modified accordingly.

p9 l8 : Could you define the Ân´ amount of lost variance Âz˙ ? Is it the difference of variance between the climatic virtual core and the intermittency virtual core ? Or the variance of a core which is reduced with increasing frequencies ?

> We include a definition of the amount of lost variance as suggested by the reviewer (lines 235 to 236):
>
> "the amount of lost variance (the difference between the variances of the intermittent and the climatic virtual cores) is positively correlated"

p9 l9 : please specify that the warm bias comes from ERA, and that it is emphasized on the plateau, thus referring to a more comprehensive study as : Fréville, H., Brun, E., Picard, G., Tatarinova, N., Arnaud, L., Lanconelli, C., ... & Van den Broeke, M. (2014). Using MODIS land surface temperatures and the Crocus snow model to un- derstand the warm bias of ERA-Interim reanalyses at the surface in Antarctica. The Cryosphere, 8(4), 1361-1373. If you rather report to the higher amount of precipitation during summer compared to winter, you should not speak about bias. Please make it clear. But in all cases consider my previous comment, at least of the ERA description.

> There seems to be a misunderstanding. We are neither talking about a bias from ERA interim, nor about a higher amount of precipitation during summer compared to winter. Instead we discuss the fact that synoptic events that bring most of the accumulation on the East Antarctic Plateau during winter have warmer conditions than the average winter conditions and thus the coldest temperatures are underrepresented in the ice-core record. This was demonstrated in earlier work e.g. Noone et al (1999) or Casado et al (2018), and can be seen in Fig. A1. We cannot discuss the biases of ERA-interim as both virtual cores are produced with ERA-interim, so if there was a warm bias in ERA-interim, we would not be able to see it. This sentence was modified to clarify this (lines 232 to 234):
>
> "These effects are included in the precipitation intermittency virtual core (Fig. \ref{fig_model_illu} (e)). Here, the amount of variance is reduced as in winter, precipitation events are often associated to warmer than average conditions, which leads to an under-representation of the coldest conditions and a warm bias in the isotopic record \citep{Noone1999,Casado2018}"

p9 l11 : Is this correlation significant ?

We included "p < 0.05".

p9 l12 : r2=0.34 is not that high, so I would rather suggest a part explanation of lost variance due to precipitation intermittency.

We changed the formulation (line 235-239) to clarify that parts of the variance reduction can be explained to the under-sampling of the colder conditions.

"Throughout Antarctica, the amount of lost variance (the difference between the variances of the intermittent and the climatic virtual cores) is positively correlated with the difference between the mean value of the intermittent virtual core and the climatic core ($r^2 = 0.34$, n = 12128, p < 0.05). This suggests that part of variance reduction is related to the under-sampling of the colder winter conditions (Fig. \ref{fig_supp_season})."

p9 l13 : The sentence Ân´ Nevertheless... Âz˙ is confusing because it does not refer to the correlation coefficient, but I guess it does ? If so, could you thus gather the two sentences the sentences related to the mean and range values of this correlation coefficient or give more clarity in anyway ?

Modified to make the point clearer into (Lines 240 to 243):

"Overall, the total amount of variance preserved in the intermittent virtual core ranges from 30 to 100\% of the amount of variance observed in the climatic signal. However, the PSD of the intermittent virtual core is very different from the climatic signal, and the large amount of variance at the frequency equivalent to 1 year in the climatic signal has been reduced as precipitation intermittency redistributes the very strong seasonal signal across all frequencies (Fig. \ref{fig_model_illu} (f))."

p9 l18 : Here I do not understand what you used to compute the correlation coefficient, once more probably because It is not clear to me what the amount of lost variance is.

At this point we are comparing the virtual core to the climatic input time-series. We separate this section into a different paragraph and make the subject of the correlation clearer (lines 243 to 249):

"Our modelling approach produces profiles of isotopic composition which can be plotted either against depth or against time. Analysing the depth series, we observe no correlation between the intermittent and the climatic virtual cores due to the seasonal cycles being out of phase due to interannual variations in the amount of precipitation. Analysing the time series, assuming that each layer is perfectly dated (perfect dating assumption), we obtain a correlation of $r = 0.85$ (p < 0.05) between the virtual core and the climate signal."

p9 l22 : Ân´ r=0.22 Âz˙. Previously, you gave r2 and not r. Could you check it was your intention to give r here. Is it significant ? What is r between the diffused virtual core and the intermittent virtual core ?

We were providing the r² before to show how much of the lost variance was explained by the warm bias (variance fraction, thus squared units). Here, we are discussing the effect of the intermittency and diffusion on the relationship of the virtual record and the climatic signal. As a result, we provide the value of r. We included that the p-value was below 0.05.

The correlation between the diffused virtual core and the intermittent virtual core is r=0.47. We decided not to include this value in the revised manuscript as we are unsure if/how this assists the interpretation of our results and as there is no physical comparison to be made with in situ data.

p9 l28 Ân´ the seasonal cycle clearly dominates the signal by roughly two orders of magnitude Âz˙, compared to another time scale ? Please specify.

The range has been added to the sentence (lines 260 to 261):

"As the seasonal cycle dominates the total temperature variability by roughly two orders of magnitude in the frequency range covered by ERA-interim"

p10 l11 : remove the dot before Ân´ see section Âz˙.

Modified accordingly.

p11 l6 : Even if the PSD dislpays the square of the amplitude, could you change it to the amplitude (thus in ‰, so it is more relevant for ice core people.

Interesting idea. However, we don't think that this would be possible and useful on our case. The PSD displays the variance per frequency bin, i.e. $‰^2$.m in our case. To transfer this into an amplitude, we would have to integrate across a certain frequency range and obtain a quantity for which the calculation of our timescale-dependent SNR would not be as straightforward as for the PSD. Most importantly, the term amplitude here would be misleading since the background spectrum of noise is not associated with any kind of oscillation for which an amplitude is defined.

P12 l6 : Can you give more information on the power law for the PSD, especially the way it is been found. You refer to an entire book, so it is not easy to find the proper dedicated paragraph.

Several studies have found that climate variability over a large range of time-scales can be described by a power law spectrum. It is also one of the simplest possible parametrisations as it only requires two parameters (intercept and slope) and it corresponds to a linear regression in the log-log plot. As there are many instances where this has been used to analyse climatic records (including ice cores), we thought a book was relevant here. We now included additional journal articles where similar approaches were taken (lines 296 to 299):

"One of the simplest parametrisations to describe the observed climate variability over a large range of time-scales is to assume a power law relationship for the PSD of the signal (S(f)) (Huybers and Curry, 2006; Lovejoy and Schertzer, 2013):"…

p13 l1-5 : Please add a robust justification for illustrating the power law only for β=0.6 and β=0.8.

The value of β=0.6 comes from Münch et al, 2018, which is to our knowledge, the only robust estimate of beta from isotopic records for the present period in East Antarctica. We give them as example of realistic values. Discussions about the range of values are provided in section 4, in order to make the choice of the values of beta clear for the reader, we include the following into the manuscript (lines 315 to 325):

"We generalise these results to all of Antarctica and present the time scales at which signal is preserved after precipitation intermittency has impacted the signal recorded in ice cores in Figure \ref{fig_TS_int_map}. We present the maps of time scale at which the signal will be preserved for values of $\beta = 0.2$ (predicted by GCM, small scaling of climate variability found in firn cores of the WAIS region \citep{Munch2018}), $\beta = 0.6$ (best guess from isotope data from East Antarctica over the last millennium \citep{Munch2018}), and a slightly higher values of $\beta = 0.8$ within the range of values expected for decadal-to-centennial scaling exponent \citep{Zhu2019}. For a value of $\beta = 0.6$, the time scales range from one year in coastal areas to 1000 years for special areas of the interior (e.g. Ellsworth Land and Victoria Land). For a value of $\beta = 0.8$, i.e. assuming more low frequency climate variability, the time scales are globally reduced. In both cases, the spatial pattern cannot be entirely explained by the amount of accumulation: while the low accumulation areas of the East Antarctic Plateau have large values of times scales at which the integrated $SNR = 1$, $\tau_a$ (from 10 to 500 years), the largest $\tau_a$ are found for around Ellsworth Land where the amount of accumulation is much larger (see Supplementary Material)"

In the revised version of the manuscript, we propose to add as well figures with β = 0.2, in addition of β = 0.6, and β = 0.8 to cover the whole range of reasonable assumptions for the climate variability of the last millennia.

p13 l5-8 and p15 l5-7: To support your argument, it would worth to add contours of accumulation in Fig. 5 and to comment it there.

We will add contours of accumulation in Fig. 5 to support the text, as well as a link to the figure presenting the accumulation that is in the supplementary materials.

p15 l8 : which snow pit data were used ? Could it be introduced into the data section ?

A new subsection has been added in the data and method section to describe the different snow pits.

p17 l2 : change $\hat{A}n'$ bellow $\hat{A}z$ to $\hat{A}n'$ below $\hat{A}z$.

Modified accordingly.

p17 l6 : change $\hat{A}n'$ an $\hat{A}z$ to $\hat{A}n'$ a $\hat{A}z$.

Modified accordingly.

p19 l3 add $\hat{A}n'$ as $\hat{A}z$ to $\hat{A}n'$ such $\hat{A}z$.

Modified accordingly.

Figures Figure 2 : That is a very nice plot !!!

Figure 3 : Are all simulated relationships signficant ? Please remove are for not signifi- cant relationships, are put hatches. Could you add contours for r from r=0.9 and 0.8, to point areas where the intermittency does relatively weakly affect the climatic signal ? Could you use the same colours for intermittency only, and intermittency and diffusion plots to make the comparision easier.

The purpose of this figure is to show the expected relationship between the climatic signal and the virtual 'corrupted' ice-core record. Therefore, a significance test (= testing if the correlation differs from the null hypothesis of no correlation) would not be meaningful. In other words, a correlation value of 0 (which is not significant by definition) contains the same amount of information (telling us that we expect no correlation to the climate signal) as a correlation value of one. The confidence intervals (telling the reader how certain the estimated correlation values are) are very narrow in our case as we have a very large amount of datapoints (N roughly equal to 59 000). We didn't display them here as it would be difficult to do this without showing triplicating the figures.

For the colour scheme, we agree that it is important to be able to compare the two maps easily and suggest an alternative solution in order to keep all the colours of the manuscript consistent (green: intermittent virtual core, blue: diffused virtual core, yellow: additional noise (probably linked to stratigraphic noise) and diffused virtual core): we will homogenise the grey density so the grey scale are the same for both figures, as a result, a black and white version of the figure would have exactly the same colour.

Figure 4 : could you add a vertical line for the seasonal frequency so it could illustrate better lines 6-7 in the text above the figure. Also detail in the legend that these fits and outputs correspond the the EDML site.

The line for the transition from seasonal to interannual is already present, but as it's perfectly aligned with the seasonal peak, it's difficult to see it. We will change the top x-axis to "1 year", "10 years", "100 years", and "1000 years" to make this clearer.

We include the needed legend that this figure was specific to the site of EDML:

> "Comparison of the amount of noise generated by precipitation intermittency and different hypothesis for the climatic signal for the site of EDML"

Figure 5 : see the specific comment suggesting adding coutours of accumulation.

> Modified accordingly.

Figure 6 and 7 : please complete the description presently suspended Ân´ have affected the signal... Âz˙.

> Modified accordingly.

Tables Table 4 : I do not see where table 4 has been cited in the text. It shows that the results are very $\beta$-dependant, and that it would have been expected similar fits than those displayed on Figure 4, in order to point the $\beta$ value to consider, and associated $\tau$ for each site.

> We included the citation to table 4 where it is needed:

> "Using spectral methods, we could determine the lower limit for the time scales at which the ice core signal is correlated with the climatic signal (Table 4)."

Casado et al. develop a model to understand the timescale on which traditional water isotopes (dD and d18O) reflect climate. They assess two processes: diffusion and intermittency of precipitation. The primary result is a lower bound for the timescale at which climate variability can be reconstructed, which ranges from a year on the coast to a millennium in the interior. The manuscript is well written and the topic is of interest as the ability to measure water isotopes at cm resolution is becoming common. The underlying modeling is sound with a strong emphasis of spectral techniques. The results of what timescale of climatic information ice cores will contain are somewhat underwhelming, since they vary by two orders of magnitude based on the assumed exponent for the power law representing long timescale climate variability. However, the primary interest of this work is not the conclusion, but rather the method development for a more quantitative understanding of processes limiting climate interpretation.

> We agree with the reviewer that parts of our results strongly depend on the assumed climate variability on decadal to millennial scales (represented here by the exponent beta) which is unfortunately poorly constrained. We will therefore improve the description of this assumption and its implications in the revised manuscript. We also note that the first part of the conclusions (relation of the climate signal and virtual cores, Figure 3) does not yet include any assumptions of climate variability and is only based on the reanalysis data. Further, even if the presented range of time-scales is large due to the missing knowledge on Antarctic climate variability, it provides bounds on the time-scales that can be reconstructed. As an example, Figure 6 shows that even assuming a strong scaling of climate variability of beta = 0.8, a value can be seen as an upper realistic bound, several regions only reach an SNR of 1 on timescales of centuries and longer.

The assumption of the power law variability and the appropriate exponent does feel neglected in the paper. The authors use 1000-year runs of GCMs to estimate the climate variability, then seem to not like the result, so basically punt on the issue and use a range from 0 to 1, although the discussion primarily focuses on 0.6 (from a single study at EDML but seems to ignore the differences with WAIS in that same study) and 0.8. Someone reading this paper quickly would think that the appropriate range would be 0.6 to 0.8 because in three different figures, these are the only values plotted. A much fuller discussion of the appropriate value needs to be included so that the reader can understand not just the impact, but the state of knowledge. A single reference to one of the group's previous papers in CP is not enough for an assumption that dominates the results.

> We agree with the reviewer that if would be valuable for the reader to add more details on the assumption of the power law variability and the appropriate exponent and that it would be useful to consider the full range of reasonable exponents instead of focusing too much on the author's best guess of beta=0.6.

> We therefore propose the following changes in the manuscript:

> - Longer discussion on the reasonable range, and why the beta can't be just estimated from single ice-cores
> - Include beta=0.2 in the figures

> For example, before Figure 5, we included more details on the range of beta value used, and why (lines 315 to 322):

> "We generalise these results to all of Antarctica and present the time scales at which signal is preserved after precipitation intermittency has impacted the signal recorded in ice cores in Figure \ref{fig_TS_int_map}. We present the maps of time scale at which the signal will be preserved for values of $\beta = 0.2$ (predicted by GCM, small scaling of climate variability found in firn cores of the WAIS region \citep{Munch2018}), $\beta = 0.6$ (best guess from isotope data from East Antarctica over the last millennium \citep{Munch2018}), and a slightly higher values of $\beta = 0.8$ within the range of values expected for decadal-to-centennial scaling exponent \citep{Zhu2019}. For a

value of $\beta = 0.6$, the time scales range from one year in coastal areas to 1000 years for special areas of the interior (e.g. Ellsworth Land and Victoria Land). For a value of $\beta = 0.8$, i.e. assuming more low frequency climate variability, the time scales are globally reduced."

Then, we included a deeper discussion about this aspect in the manuscript, referencing papers from other groups with relevant results (lines 496 to 510):

"Finally, another important aspect that limits the ability of our modelling approach to provide time scales at which an actual ice core can be used is the lack of constraints on the strength of decadal to centennial climate variability in Antarctica and at the ice core sites, reflected here by the values of $\beta$. We were not able to estimate the scaling for each site and instead presented results for values of $\beta$ of 0.2, 0.6 and 0.8 to cover the whole range of reasonable assumptions for the climate variability of the last millennia. The lowest value of $\beta$ of 0.2 represents the variability as simulated from current climate models but is likely pessimistic as current climate models tend to underestimate regional climate variability \citep{Laepple2014}. It further reflects the small scaling of climate variability that was found in firn-cores of the WAIS region (\citep{Munch2018}). The value of $\beta = 0.6$ represents our best guess based on a single study that estimated the decadal-to-centennial scaling exponents from an array of ice core records in in the EDML region (\citep{Munch2018}). Finally, the highest value $\beta$ of 0.8 represents an optimistic assumption of strong slow climate variability, similar to estimates for regional ocean variability of $\beta = 1$ \citep{Laepple2014}. Our choice is based on the reasoning that while on interannual to decadal time-scales climate variability over land scales less than over the ocean \citep{Huybers2006}, they should converge to a similar scaling behaviour on longer-time scales \citep{North2011}. Further analyses of firn and ice-core arrays, paired with a better understanding of the noise processes such as presented here, would provide a mean to make significant progress in the interpretation of ice core records and Antarctic climate variability."

The forward model also seems to neglect two important processes: clear-sky precipitation and atmosphere-surface snow isotopic exchange. Both of these have the potential to offset the effect of precipitation intermittency. Assuming ERAi captures the timing of precipitation is fine in high precipitation areas, but these are also of the least interest since they receive precipitation relatively consistently. This assumption is much less valid for central East Antarctica, where clear-sky precipitation can make up a significant portion of total precipitation due to the infrequency and low volume of precipitation events. As to atmosphere-surface snow exchange, quantitative modeling would be challenging given the relative lack of information on the topic, but a qualitative discussion of its effect would be valuable.

We included in depth discussion about both these processes. First, for clear sky precipitation, we included a sensitivity test in which 50% of the accumulation originated from clear sky precipitation, and only half was generated by the long range precipitation (main type of precipitation generated by ERA-interim). This results into this addition into the discussion (lines 455 to 470):

"Second, we did not include the effect of clear sky precipitation, which, in the center of Antarctica can sum up to a significant fraction of the annual accumulation \citep{Fujita2006} and would tend to occur more evenly over the year, in contrast with the intermittent aspect of long-range precipitation (the only one included accurately in ERA-interim). Such a more regular input of the signal would reduce the amount of noise created by precipitation intermittency. Sensitivity tests show that if half of the precipitation originates from clear sky precipitation, the time scales at which the signal can be retrieved after precipitation intermittency is significantly larger (for $\beta = 0.6$, at EDML, 3.4 years instead of 17, see Supplementary Material S4). However, as the relationship between clear sky precipitation isotopic composition and temperature likely differs from the one between long range precipitation and temperature \citep{Dittmann2016,Stenni}, we expect additional noise from mixing these two different signal sources of signal which would decrease the time scale at which the signal can be retrieved (3.9 years, see Supplementary Material S4). These limitations are overall linked to uncertainties in the surface mass balance \citep{Genthon2015,Picard2019}, which in Central Antarctica are quite large, in particular due to analytical limits to evaluate the precipitation amounts (both clear sky and long range),

the snow blown and redeposited by the wind \citep{Palm2017}, and sublimation and condensation \citep{Genthon2017}. In Section \ref{sec_stratnoise}, we showed that the total noise level appeared to be higher than what was predicted by just precipitation intermittency, which would suggest that the reduced white noise level generated by precipitation intermittency including clear sky precipitation would have to be compensated by additional noise needed after precipitation intermittency (provided that we know the total noise level)."

And these additions to the supplementary materials:

4.    Addition of clear sky precipitation

In dry areas of the East Antarctic Plateau, it is often observed that precipitation occurs without any cloud cover. This precipitation, known as clear sky precipitation, can account for up to 50% of the total amount (Fujita et al, 2006). This regular input of small amount of precipitation every day would tend to reduce the impact of precipitation intermittency on the ice core records. This type of precipitation is not included in ERA-interim, so to test the impact it would have on the time scales at which the signal is affected by precipitation intermittency, we computed the forward model assuming that half of the accumulation originates from clear sky precipitation. To do so, we divided by two the precipitation variable and added a constant precipitation input everyday balancing the removed precipitation. Furthermore, we assumed that the link between isotopic composition and temperature is the same for both types of precipitation.

For the EDML site, the noise level added by precipitation intermittency including clear sky precipitation is $0.21‰^2$m, i.e. almost three times below the value obtained when all of the accumulation originates from long range precipitation, which is more intermittent ($0.59‰^2$m). As a result, the time scales at which the signal can be retrieved are three to ten times lower taking into account a regular input of clear sky precipitation (Table S1).

**Table S2:** Time scales $\tau_a$ for which $SNR = 1$ for averaged samples for precipitation intermittency at EDML, including clear sky precipitation with the same isotopic signature than long range precipitation.

| $\beta$ | 0 | 0.2 | 0.4 | 0.6 | 0.8 | 1 |
|---|---|---|---|---|---|---|
| $\tau_a(a)$ | / | 56 | 7.1 | 3.4 | 2.1 | 1.4 |

Clear sky precipitation is usually following a very different distillation path than the one experienced by long range precipitation, and the associated isotopic sensitivity to temperature is also expected to be different (Dittman et al, 2016, Stenni et al, 2016). We implemented this by attributing a different relationship for the link between isotopic composition of clear sky precipitation and temperature ($0.51‰.C^{-1}$) than for the link between isotopic composition of long range precipitation and temperature ($0.41‰.C^{-1}$). In this case, the amount of noise created by the precipitation intermittency is slightly higher ($0.23‰^2$m) than in the previous case for clear sky precipitation ($0.21‰^2$m), and as a result, the time scales for which the signal is preserved are higher (Table S3). This increase of $\tau$_a is very small compared to the initial decrease due to the inclusion of clear sky precipitation, suggesting that clear sky precipitation would tend in either case to limit the impact of precipitation intermittency on the time scale at which signal is preserved in ice core records.

**Table S3:** Time scales $\tau_a$ for which $SNR = 1$ for averaged samples for precipitation intermittency at EDML, including clear sky precipitation with a different isotopic signature ($0.51‰.C^{-1}$) than long range precipitation ($0.41‰.C^{-1}$).

| $\beta$ | 0 | 0.2 | 0.4 | 0.6 | 0.8 | 1 |
|---|---|---|---|---|---|---|
| $\tau_a(a)$ | / | 91 | 8.8 | 3.9 | 2.4 | 1.5 |

Second, for atmosphere-surface snow exchanges, we expect these changes to be affected by the stratigraphy as latent fluxes are driven by the radiative heating of the surface, for which surface dune structure will be a key factor. At the time scales at hand here, this signal will be globally mixed with the wind-blown snow input and difficult to use to parameterise the stratigraphic noise with the present level of data we have. We included this discussion into the main text (lines 471 to 479):

"Third, exchanges between the surface snow and the atmosphere can lead to significant changes of the snow isotopic composition in between precipitation events in polar regions \citep{Steen-Larsen2014,Ritter2016,Casado2018}. These effects are not directly taken into account here, and would take part after the precipitation has been deposited, while the snow remains at the surface \citep{Munch2017}. On one hand, snow-atmosphere exchanges would be more continuous over time and thus reduce the noise level in case that these effects play a significant role compared to the isotopic variations driven by the isotopic content of the snow-fall. On the other hand, sublimation and condensation could result in an addition of noise at the surface due to the strong spatial (linked to dune and sastrugi) and temporal (linked to variable cloud cover) variability in latent heat created mostly by radiative processes \citep{Vignon2017a}. }"

Overall, this paper represents useful work that with a few improvements will contribute to water isotope analysis in ice cores.

Specific comments: This manuscript does not have line numbers, which should be remedied for future CP discussion papers because it makes referring to specific in- stances very difficult for both the referee and the authors.

This issue with the latex compilation of the manuscript seems to be fixed now. I apologise to the reviewers for the inconvenience this created.

- physico-chemical > why create this abomination of a hyphenation? Just write it out

Modified accordingly.

- P2, last paragraph, consider using duration instead of size qualifiers, so replace larger with longer when referring to a timescale

Modified accordingly.

- P2, last paragraph, change "read" to "interpret"

Modified accordingly.

- P2, last paragraph, I think "virtually unlimited" is an overstatement given dispersion in CFA systems is still a limiting factor in the effective resolution

Modified to "Although technical advances in analytical techniques \citep{Jones2017a} have immensely improved the sampling resolution without any additional analytical costs, there is still a need" (lines 55 to 57)

- P4, third paragraph, I'm really confused about what is being said in this paraph about perfect dating. I think I follow that because you tag the age of layers in the virtual core, you can compare these without age uncertainty to the original climate signal. However, you don't explain how you then stretch between the tagged ages. And the discussion of the depth scale getting out of phase seems both obvious and also not relevant, adding confusion.

We added more details on the perfect dating process. This is a very technical aspect we use the spectra in the depth domain to avoid the dating issue throughout the manuscript. We illustrate this point for the comparison of the correlation in the time domain (maps for Fig. 3). The discussion of the depth scale

getting out of phase seems quite relevant to us, as this is the reason why we have to do the perfect dating. It is indeed obvious to us, but we are not sure it would be for everyone. We hope that the modified version would add clarity (lines 108 to 119):

"The virtual records (intermittent virtual core and diffused virtual core) are block-averaged to create a 1\unit{cm} vertical resolution, similar to what can be achieved with manual sampling of ice cores. The virtual ice cores are perfectly dated by tagging the formation date and time to each layer. During the block-averaging to 1\unit{cm}, we also block-average the date tags to obtain the average age of each 1\unit{cm} layer. This perfect dating is used to compare the original climatic signal to the generated virtual cores, in an optimistic case (without age uncertainty). To do so, we do a linear interpolation of the virtual ice core to the original date coordinates of the input data. Indeed, the original climatic depth series typically shows a rather poor correlation with the generated virtual cores as their respective depth axes move quickly out of phase due to the large interannual variability in precipitation, which creates years accounting for thicker/thinner layers when the amount of precipitation is large/small. In contrast, a perfect record would only contain the climatic signal and produce the same layer thickness each year. The perfect dating enables to synchronise the virtual cores' time series on the climatic signal in order to provide an upper bound of how meaningful a reconstruction would be ignoring dating issues."

- P4, last paragraph. This paragraph misrepresents the impacts of wind scour. The references (i.e. Picard et al., 2019) appear to be focused solely on interior East Antarc- tica, whereas the authors here imply that Antarctica as a whole can lose 90% of its accumulation to wind scour, thus explaining the ERAi overestimation of total precipitation. The authors need to be specific about when they are writing about Antarctica as a whole, and when only specific regions. ERAi is well known to get precipitation timing correct, but magnitudes off, with Medley et al. 2013, GRL being a good reference.

We included more precision on the area that the reference is valid for the statement to wind scour (lines 124 to 134):

"The ERA-interim temperature data provide good approximations of the spatial and temporal variations of the temperature observed in \textit{in-situ} data from Antarctica {\color{red} \citep{Genthon2013,Medley2013,Jones2015}}. However, compared to satellite products and \textit{in-situ} ice core records, the ERA interim data overestimates the amounts of total precipitation by 50 to 95\% \citep{Arthern2006,Thomas2017}. This is to be expected as precipitation accounts only for a fraction of accumulation in Antarctica, especially in Central Antarctica, where up to 90\% of the local accumulation can be blown away by wind \citep{Picard2019} and more than 10\% of the total surface mass balance can be associated to sublimation and condensation \citep{Genthon2017}. Nevertheless, precipitation occurrence tends to correlate well with \textit{in-situ} snow fall events in the interior of Antarctica \citep{Medley2013,Libois2014}, and with ice core records from Antarctica \citep{Sime2011}. This supports the use of ERA-interim precipitation since a well-captured precipitation variability is needed to realistically model the precipitation intermittency."

We included the reference mentioned by the reviewer.

- P5, first full paragraph, how can the satellite data of Arthern 2006 have been corrected to the data in a 2017 paper. Be clear with the reference. Either cite older sources, or be specific about what data in the Thomas paper was used.

We included more details about this description to explain how we use the data from the Thomas paper to correct the data from Arthern (lines 135 to 149):

"However, since the diffusion length depends on the amount of accumulation, we need to compensate for the difference between precipitation and accumulation. This is achieved by applying an individual linear correction at each grid point of the reanalysis product. The correction matrix was generated using satellite data of snow accumulation \citep{Arthern2006}, that were preliminary corrected to match the accumulation obtained in ice core records \citep{Thomas2017} with the following procedure. For the

virtual cores to have the same accumulation as actual ice cores, we use a reference accumulation rate for the years from 1960 to 2016 from a recently established database of regional Antarctic snow accumulation from ice core records over the past 1000 years \citep{Thomas2017}. We selected all the ice cores sites with accumulations ranging from 20 to 400\unit{kg \: m^{-2} \: a^{-1}} that had overlap with the ERA-interim time series (in total 71). The accumulation range upper limit (400\unit{kg \: m^{-2} \: a^{-1}}) was chosen to be representative of the low accumulation rates of the deep ice core sites (in general <100\unit{kg \: m^{-2} \: a^{-1}}) where the results are more sensitive to the use of an accurate accumulation rate. We then did a spatial linear regression between the satellite derived accumulation \citep{Arthern2006} for these 71 sites and the ice cores observations, and use the produced regression to calibrate the satellite data of snow accumulation on the ice core accumulation rates. Finally, we interpolated the corrected satellite product to ERA-interim grid, and use the corrected satellite product as a reference for the accumulation, normalising the precipitation amount of ERA-interim to match this reference."

- P6 – define SNR in an equation. It gets confusing because you use SNR in an C3 equation before your present an alternate SNR(f) in a subsequent equation.

Modified accordingly. The changes, also responding to the first reviewer's suggestion, include a description of each tool before we start using them. Changes include (Lines 174 to 195):

"To evaluate the extent to which the climatic signal is preserved in the ice core record as a function of time scale, we assess the minimum time scales $\tau$ at which the SNR reaches a value of $1$. Since in any proxy record containing a climatic signal and noise, the SNR and the correlation between the record and the climatic signal are linked via:

\begin{equation}

r^2 = \frac{SNR}{1 + SNR}

\end{equation}

As a result, this minimum time scale will correspond to a correlation with the climate signal of $r = \sqrt{0.5}\sim0.71$.

\newline

The SNR can be defined in two ways. First, one can analyse the SNR at a specific frequency after filtering the ice-core time series with a narrow bandpass filter, in which case we refer to the minimum time scale as $\tau_b$, where the subscript $b$ stands for bandpass. Second, and more commonly, the ice-core time series results from averaging a higher-resolution record to a fixed temporal resolution, either by discrete sampling in the depth domain or by averaging the dated record to a specific resolution, e.g. annual or decadal resolution. In this case, we refer to the minimum time scale as $\tau_a$, where the subscript $a$ stands for averaging.

\newline

Formally, $\tau_b$ is given by the critical frequency $f_b=1/\tau_b$, for which the direct ratio of the signal and noise spectra reaches $1$,

\begin{equation}

 SNR(f_b) = \mathcal{S}(f_b)/\mathcal{N}(f_b) = 1

\end{equation}

where $\mathcal{S}$ and $\mathcal{N}$ are the PSDs of the signal and the noise. To obtain $tau\_a$, the PSDs of the signal and noise have to be integrated to that critical frequency $f\_a=1/tau\_a$, for which the ratio of the \emph{integrated} spectra reaches $1$,

\begin{equation}

SNR(f) = \frac{\int_{1/L_{R}}^{f_a} \mathcal{S}(\nu) d\nu}{\int_{1/L_{R}}^{f_a} \mathcal{N}(\nu) d\nu} = 1

\end{equation}

where $L_R$ is the length of the record (either in years or in metres). Graphically, this is given by the ratio of the area representing the signal excess $\mathcal{A}_{signal}$ to the area representing the noise excess $\mathcal{A}_{noise}$ (Fig. 1a). If the signal is redder than the noise, the minimum time scale $tau\_a$ will be smaller than $tau\_b$.}"

- Figure 2: This is a well crafted figure that illustrates the methods well

> Thanks.

- Figure 4: This figure is hard to interpret. There seems to be about every shade and thickness of red line possible. Differentiate the climate model coloring. Describe the beta=0.2 fit. Label each of the power law exponent lines on the left, rather than just varying the thickness. Or maybe you want to rethink this figure all together and break it into parts, more like Figure 2.

> We will propose an alternative version of the figure using the suggestions of the reviewer.

- Combine Tables 1 and 2 and use the caption as an opportunity to distinguish what tao_a and tao_b are.

> Modified accordingly.

- Figure 5, the use of just 0.6 and 0.8 feels misleading given the uncertainty in this parameter choice. At the least, I think beta=0.2 needs to be included, since it will show up in table 4. Same goes for Figures 6 and 9. Also, all these figures should have the ice core locations of sites in Table 4 shown.

> We agree with the reviewer, and will include β = 0.2 in Figures 5, 6 and 9 as well as the ice core locations. In depth discussions have been given for the choice of the values of beta illustrated here, in particular ahead of Figure 5 (lines 315 to325):

> ""We generalise these results to all of Antarctica and present the time scales at which signal is preserved after precipitation intermittency has impacted the signal recorded in ice cores in Figure \ref{fig_TS_int_map}. We present the maps of time scale at which the signal will be preserved for values of $\beta = 0.2$ (predicted by GCM, small scaling of climate variability found in firn cores of the WAIS region \citep{Munch2018}), $\beta = 0.6$ (best guess from isotope data from East Antarctica over the last millennium \citep{Munch2018}), and a slightly higher values of $\beta = 0.8$ within the range of values expected for decadal-to-centennial scaling exponent \citep{Zhu2019}. For a value of $\beta = 0.6$, the time scales range from one year in coastal areas to 1000 years for special areas of the interior (e.g. Ellsworth Land and Victoria Land). For a value of $\beta = 0.8$, i.e. assuming more low frequency climate variability, the time scales are globally reduced. In both cases, the spatial pattern cannot be entirely explained by the amount of accumulation (Fig. 6 and Fig. S1) while the low accumulation areas of the East Antarctic Plateau have large values of times scales at which the integrated $SNR = 1$, $\tau\_a$ (from 10 to 500 years), the largest $\tau\_a$ are found for around Ellsworth Land where the amount of accumulation is much larger (see Supplementary Material)"

We will include the ice core locations that are given in Table 4.

- Table 3: I'm not sure what tao without a subscript is. Is this tao_a like in the caption?

Yes, modified accordingly

- P17, "below" not "bellow", but I would like to see "bellow" used in a CP paper sometime

Modified accordingly

- P20: section 4.3, I appreciate the inclusion of a section like this, but it seems like many limits of the approach were not discussed. Expand this with discussion of beta and atmosphere-surface snow isotopic exchange.

We include more details in the section 4.3 (lines 435 to 510):

[revised manuscript text omitted]

Finally, another important aspect that limits the ability of our modelling approach to provide time scales at which an actual ice core can be used is the lack of constraints on the strength of decadal to centennial climate variability in Antarctica and at the ice core sites, reflected here by the values of $\beta$. We were not able to estimate the scaling for each site and instead presented results for values of $\beta$ of 0.2, 0.6 and 0.8 to cover the whole range of reasonable assumptions for the climate variability of the last millennia. The lowest value of $\beta$ of 0.2 represents the variability as simulated from current climate models but is likely pessimistic as current climate models tend to underestimate regional climate variability \citep{Laepple2014}. It further reflects the small scaling of climate variability that was found in firn-cores of the WAIS region (\citep{Munch2018}). The value of $\beta = 0.6$ represents our best guess based on a single study that estimated the decadal-to-centennial scaling exponents from an array of ice core records in in the EDML region (\citep{Munch2018}). Finally, the highest value $\beta$ of 0.8 represents an optimistic assumption of strong slow climate variability, similar to estimates for regional ocean variability of $\beta = 1$ \citep{Laepple2014}. Our choice is based on the reasoning that while on interannual to decadal time-scales climate variability over land scales less than over the ocean \citep{Huybers2006}, they should converge to a similar scaling behaviour on longer-time scales \citep{North2011}. Further analyses of firn and ice-core arrays, paired with a better understanding of the noise processes such as presented here, would provide a mean to make significant progress in the interpretation of ice core records and Antarctic climate variability.